# Feature learning via mean-field Langevin dynamics: classifying sparse parities and beyond

**Taiji Suzuki**[1,2], **Denny Wu**[3,4], **Kazusato Oko**[1,2], **Atsushi Nitanda**[2,5]

[1]University of Tokyo,  [2]RIKEN AIP,  [3]New York University,
[4]Flatiron Institute,  [5]Kyushu Institute of Technology

`taiji@mist.i.u-tokyo.ac.jp, dennywu@nyu.edu,`
`oko-kazusato@g.ecc.u-tokyo.ac.jp, nitanda@ai.kyutech.ac.jp`

## Abstract

Neural network in the mean-field regime is known to be capable of *feature learning*, unlike the kernel (NTK) counterpart. Recent works have shown that mean-field neural networks can be globally optimized by a noisy gradient descent update termed the *mean-field Langevin dynamics* (MFLD). However, all existing guarantees for MFLD only considered the *optimization* efficiency, and it is unclear if this algorithm leads to improved *generalization* performance and sample complexity due to the presence of feature learning. To fill this important gap, in this work we study the sample complexity of MFLD in learning a class of binary classification problems. Unlike existing margin bounds for neural networks, we avoid the typical norm control by utilizing the perspective that MFLD optimizes the *distribution* of parameters rather than the parameter itself; this leads to an improved analysis of the sample complexity and convergence rate. We apply our general framework to the learning of $k$-sparse parity functions, where we prove that unlike kernel methods, two-layer neural networks optimized by MFLD achieves a sample complexity where the degree $k$ is "decoupled" from the exponent in the dimension dependence.

## 1   Introduction

**Mean-field Langevin dynamics.**   The optimization dynamics of two-layer neural networks in the mean-field regime can be described by a nonlinear partial differential equation of the distribution of parameters (Nitanda and Suzuki, 2017; Chizat and Bach, 2018; Mei et al., 2018; Rotskoff and Vanden-Eijnden, 2018; Sirignano and Spiliopoulos, 2020). Such a description has multiple advantages: $(i)$ global convergence guarantees can be obtained by exploiting convexity of the loss function, and $(ii)$ the parameters are allowed to evolve away from initialization and learn informative features, in contrast to the *neural tangent kernel* ("lazy") regime (Jacot et al., 2018).

Among the gradient-based optimization algorithms for mean-field neural networks, the *mean-field Langevin dynamics* (MFLD) (Mei et al., 2018; Hu et al., 2019) is particularly attractive due to the recently established *quantitative* optimization guarantees. MFLD arises from a noisy gradient descent update on the parameters, where Gaussian noise is injected to the gradient to encourage "exploration". It has been shown that MFLD globally optimizes an *entropy-regularized convex functional* in the space of measures, and for the infinite-width and continuous-time dynamics, the convergence rate is exponential under suitable isoperimetric conditions (Nitanda et al., 2022; Chizat, 2022). Furthermore, uniform-in-time estimates of the particle discretization error have also been established (Suzuki et al., 2023a; Chen et al., 2022), meaning that optimization guarantees for the infinite-dimensional problem can be effectively translated to a finite-width neural network.

However, existing analyses of MFLD only considered the *optimization* of neural networks; this alone does not demonstrate the benefit of mean-field regime nor the presence of feature learning. Therefore,

an important problem is to characterize the *generalization* of the learned models, and prove efficient sample complexity guarantees. The goal of this work is to address the following question.

*Can we show that neural network + simple noisy gradient descent (MFLD) efficiently learns an interesting class of functions, with a better rate of convergence compared to the "lazy" regime?*

**Learning sparse parity functions.** One particularly relevant learning task is the $k$-sparse parity problem, where the response $y$ is given by the sign of the product of $k$ coordinates of the input (on hypercube); as a special case, setting $k = 2$ recovers the classical XOR problem. When $k \ll d$, this target function is low-dimensional, and hence we expect feature learning to be beneficial in that it can "zoom in" to relevant subspace. In contrast, for kernel methods (including neural networks in the lazy regime) which cannot adapt to such structure, it has been shown that a sample complexity of $n = \Omega(d^k)$ is unavoidable (Ghorbani et al., 2019; Hsu; Abbe et al., 2022).

For the XOR case ($k = 2$), recent works have shown that neural networks in the mean-field regime can achieve a sample complexity of $n = \mathcal{O}(d/\epsilon)$ (Wei et al., 2019; Chizat and Bach, 2020; Telgarsky, 2023), which indeed improves upon the NTK complexity (Ji and Telgarsky, 2019). However, all these results directly assumed convergence of the dynamics ($t \to \infty$) with no iteration complexity. Moreover, Wei et al. (2019); Chizat and Bach (2020) directly analyzed the infinite-width limit, and while Telgarsky (2023) provided a finite-width characterization, the dynamics is restricted to the low-rotation regime, and a very large number of particles $N = \mathcal{O}(d^d)$ is required. Lastly, these analyses are specialized to XOR, and do not directly generalize to the $k$-parity setting.

## 1.1 Our Contributions

In this work, we bridge the aforementioned gap by presenting a simple and general framework to establish sample complexity of MFLD in learning binary classification problems. We then apply this framework to the sparse $k$-parity problem, and obtain improved rate of convergence for the fully time- and space-discretized algorithm. More specifically, our contributions can be summarized as follows.

- We present a general framework to analyze MFLD in the learning of binary classification tasks. Our framework has two main ingredients: ($i$) an annealing procedure that applies to common classification losses that removes the exponential dependence on regularization parameters in the *logarithmic Sobolev inequality*, and ($ii$) a novel local Rademacher complexity analysis for the distribution of parameters optimized by MFLD. As a result, we can obtain generalization guarantee for the learned neural network in discrete-time and finite-width settings.

- We apply our general framework to the $k$-sparse parity problem, and derived learning guarantees with improved rate of convergence and dimension dependence, as shown in Table 1. Specially, in the $n \asymp d^2$ regime we obtain exponentially converging classification error, whereas in the $n \asymp d$ regime we achieve linear dimension dependence. Note that this improves upon the NTK analysis (which gives a sample complexity of $n = \Omega(d^k)$) in that it "decouple" the degree $k$ from the exponent in the dimension dependence. Our theoretical results are supported by empirical findings.

| Authors | regime/method | $k$-parity | class error | width | # iterations |
|---|---|---|---|---|---|
| Ji and Telgarsky (2019) | NTK/SGD | No | $d^2/n$ | $d^8$ | $d^2/\epsilon$ |
| Telgarsky (2023) | NTK/SGD | No | $d^2/n$ | $d^2$ | $d^2/\epsilon$ |
| Barak et al. (2022) | Two phase SGD | Yes | $d^{(k+1)/2}/\sqrt{n}$ | $O(1)$ | $d/\epsilon^2$ |
| Wei et al. (2019) | mean-field/GF | No | $d/n$ | $\infty$ | $\infty$ |
| Telgarsky (2023) | mean-field/GF | No | $d/n$ | $d^d$ | $\infty$ |
| Ours | mean-field/MFLD | Yes | $\exp(-O(\sqrt{n}/d))$ | $e^{O(d)}$ | $e^{O(d)}$ |
| Ours | mean-field/MFLD | Yes | $d/n$ | $e^{O(d)}$ | $e^{O(d)}$ |

Table 1: Statistical and computational complexity (omitting poly-log terms) for the $k$-sparse (or 2-sparse) parity problem. Column "$k$-parity" indicates applicability to the general $k$-sparse parity setting, and for references that does not handle $k$-parity we state the complexity for 2-parity (XOR). $d$ is the input dimensionality, and $n$ is the sample size. For SGD, $n$ is the total sample size given by product of mini-batch size and iterations.

## 1.2 Additional Related Works

In addition to the mean-field analysis, parity-like functions can be learned via other feature learning procedures such as the one gradient step analysis in Daniely and Malach (2020); Ba et al. (2022);

Damian et al. (2022); Barak et al. (2022). Such analysis requires nontrivial gradient concentration at initialization, which translates to a sample complexity that scales as $n = \Theta(d^k)$ (Barak et al., 2022). For "narrow" neural networks, Abbe et al. (2023) showed that a modified projected (online) gradient descent algorithm can learn the $k$-parity problem with a sample complexity of $n = O(d^{k-1} + \epsilon^{-c})$ (for some unknown $c$). Also, Refinetti et al. (2021); Ben Arous et al. (2022) showed that a two-layer ReLU network with more than 4 neurons can learn the Gaussian XOR problem with gradient descent.

## 2   Problem Setting

Throughout this paper, we consider a classification problem given by the following model:

$$Y = \mathbf{1}_A(Z) - \mathbf{1}_{A^c}(Z) \in \{\pm 1\}$$

where $Z = (Z_1, \ldots, Z_d)$ is the input random variable on $\mathbb{R}^d$ and $\mathbf{1}_A$ is the indicator function corresponding to a measurable set $A \in \mathcal{B}(\mathbb{R}^d)$, i.e., $\mathbf{1}_A(Z) = 1$ if $Z \in A$ and $\mathbf{1}_A(Z) = 0$ if $Z \notin A$. Let $P_Z$ be the distribution of $Z$. We are given input-output pairs $D_n = (z_i, y_i)_{i=1}^n$ independently identically distributed from this model as training data. Then, we construct a binary classifier that predicts the label for the test input data as accurate as possible. To achieve this, we learn a two-layer neural network model in the mean-field regime via the *mean-field Langevin dynamics*.

One important problem setting for our analysis is the $k$-sparse parity problem defined as follows.

**Example 1** ($k$-sparse parity problem). *$P_Z$ is the uniform distribution on the grid $\{\pm 1/\sqrt{d}\}^d$ and $A = \{\zeta = (\zeta_1, \ldots, \zeta_d) \in \{\pm 1/\sqrt{d}\}^d \mid \zeta_1 \cdots \zeta_k > 0\}$[1].*

As a special case, $k = 2$ (XOR) has been extensively studied (Wei et al., 2019; Telgarsky, 2023).

**Mean-field two-layer network.**   Given input $z$, let $h_x(z)$ be one neuron in a two-layer neural network with parameter $x = (x_1, x_2, x_3) \in \mathbb{R}^{d+1+1}$ defined as

$$h_x(z) = \bar{R}[\tanh(z^\top x_1 + x_2) + 2\tanh(x_3)]/3,$$

where $\bar{R} \in \mathbb{R}$ is a hyper-parameter determining the scale of the network[2]. We place an extra $\tanh$ activation for the bias term $x_3 \in \mathbb{R}$ because the boundedness of $h_x$ is required in the convergence analysis. Let $\mathcal{P}$ be the set of probability measures on $(\mathbb{R}^{\bar{d}}, \mathcal{B}(\mathbb{R}^{\bar{d}}))$ where $\bar{d} = d + 2$ and $\mathcal{B}(\mathbb{R}^{\bar{d}})$ is the Borel $\sigma$-algebra on $\mathbb{R}^{\bar{d}}$ and $\mathcal{P}_p$ be the subset of $\mathcal{P}$ such that its $p$-th moment is bounded: $\mathbb{E}_\mu[\|X\|^p] < \infty$ ($\mu \in \mathcal{P}$). The mean-field neural network is defined as an integral over neurons $h_x$,

$$f_\mu(\cdot) = \int h_x(\cdot)\mu(\mathrm{d}x),$$

for $\mu \in \mathcal{P}$. To evaluate the performance of $f_\mu$, we define the empirical risk and the population risk as

$$L(\mu) := \tfrac{1}{n}\sum_{i=1}^n \ell(y_i f_\mu(z_i)), \quad \bar{L}(\mu) := \mathbb{E}[\ell(Y f_\mu(Z))],$$

respectively, where $\ell : \mathbb{R} \to \mathbb{R}_{\geq 0}$ is a convex loss function. In particular, we consider the logistic loss $\ell(f, y) = \log(1 + \exp(-yf))$ for $y \in \{\pm 1\}$ and $f \in \mathbb{R}$. To avoid overfitting, we consider a regularized empirical risk $F(\mu) := L(\mu) + \lambda\mathbb{E}_{X \sim \mu}[\lambda_1\|X\|^2]$, where $\lambda, \lambda_1 \geq 0$ are regularization parameters. One advantage of this mean-field definition is that $f_\mu$ is a linear with respect to $\mu$, and hence the functional $L(\mu)$ becomes a convex functional.

**Mean-field Langevin dynamics.**   We optimize the training objective via MFLD, which is given by the following stochastic differential equation:

$$\mathrm{d}X_t = -\nabla\frac{\delta F(\mu_t)}{\delta\mu}(X_t)\mathrm{d}t + \sqrt{2\lambda}\mathrm{d}W_t, \quad \mu_t = \mathrm{Law}(X_t), \tag{1}$$

where $X_0 \sim \mu_0$, $\mathrm{Law}(X)$ denotes the distribution of the random variable $X$ and $(W_t)_{t \geq 0}$ is the $d$-dimensional standard Brownian motion. Readers may refer to Theorem 3.3 of Huang et al. (2021) for the existence and uniqueness of the solution. Here, $\frac{\delta F(\mu_t)}{\delta\mu}$ is the first variation of $F$.

**Definition 1.** *For a functional $G : \mathcal{P} \to \mathbb{R}$, the first-variation $\frac{\delta G}{\delta\mu}(\mu)$ at $\mu \in \mathcal{P}$ is a continuous functional $\mathcal{P} \times \mathbb{R}^d \to \mathbb{R}$ satisfying $\lim_{\epsilon \to 0}\frac{G(\epsilon\nu + (1-\epsilon)\mu)}{\epsilon} = \int\frac{\delta G}{\delta\mu}(\mu)(x)\mathrm{d}(\nu - \mu)$ for any $\nu \in \mathcal{P}$.*

---

[1]We present the axis-aligned setting for conciseness, but the same result holds under orthogonal transforms.
[2]We may also deal with more general activation function that is smooth and bounded.

In our setting, we have $\frac{\delta F(\mu)}{\delta \mu}(x) = \frac{1}{n} \sum_{i=1}^{n} \ell'(y_i f_\mu(z_i)) y_i h_x(z_i) + \lambda(\lambda_1 \|x\|^2)$. It is known that the Fokker-Planck equation of the SDE (1) is given by[3]

$$\partial_t \mu_t = \lambda \Delta \mu_t + \nabla \cdot \left[ \mu_t \nabla \frac{\delta F(\mu_t)}{\delta \nu} \right] = \nabla \cdot \left[ \mu_t \nabla \left( \lambda \log(\mu_t) + \frac{\delta F(\mu_t)}{\delta \nu} \right) \right]. \tag{2}$$

Then, we can verify that this is equivalent to the Wasserstein gradient flow to optimize the following entropy regularized risk (Mei et al., 2018; Hu et al., 2019):

$$\mathcal{L}(\mu) = F(\mu) + \lambda \mathrm{Ent}(\mu) = L(\mu) + \lambda \mathrm{KL}(\nu, \mu) + (\text{const.}) \tag{3}$$

where $\mathrm{KL}(\nu, \mu) = \int \log(\mu/\nu) \mathrm{d}\mu$ is the KL divergence between $\nu$ and $\mu$, and $\nu$ is the Gaussian distribution with mean 0 and variance $I/(2\lambda_1)$, i.e., $\nu = \mathcal{N}(0, I/(2\lambda_1))$.

For a practical algorithm, we need to consider a space- and time-discretized version of the MFLD, that is, we approximate the solution $\mu_t$ by an empirical measure $\mu_{\mathscr{X}} = \frac{1}{N} \sum_{i=1}^{N} \delta_{X_i}$ corresponding to a set of finite particles $\mathscr{X} = (X^i)_{i=1}^{N} \subset \mathbb{R}^{\bar{d}}$. Let $\mathscr{X}_\tau = (X_\tau^i)_{i=1}^{N} \subset \mathbb{R}^{\bar{d}}$ be $N$ particles at the $\tau$-th update ($\tau \in \{0, 1, 2, \dots\}$), and define $\mu_\tau = \mu_{\mathscr{X}_\tau}$ as a finite particle approximation of the population counterpart. Then, the discretized MFLD is defined as follows: $X_0^i \sim \mu_0$, and $\mathscr{X}_\tau$ is updated as

$$X_{\tau+1}^i = X_\tau^i - \eta \nabla \frac{\delta F(\mu_\tau)}{\delta \mu}(X_\tau^i) + \sqrt{2\lambda\eta} \xi_\tau^i, \tag{4}$$

where $\eta > 0$ is the step size, and $\xi_\tau^i \sim_{i.i.d.} N(0, I)$. This is the Euler-Maruyama approximation of the MFLD with a discretized measure; we present the discretization error bounds in the next section.

## 3 Main Assumptions and Theoretical Tools

In this section, we introduce the basic assumptions and technical tools for our analysis.

**Condition on the loss function** To derive the convergence of the classification error, we assume that the loss function satisfies the following condition.

**Assumption 1.** *The convex loss function $\ell : \mathbb{R} \to \mathbb{R}_{\geq 0}$ satisfies the following conditions:*

- *$\ell$ is first order differentiable, its derivative is Lipschitz continuous and its derivative is bounded by 1: $|\ell'(x) - \ell'(x')| \leq C|x - x'|$ and $\sup_x |\ell'(x)| \leq 1$.*
- *$\ell$ is monotonically decreasing, and is classification calibrated: $\ell'(0) < 0$ (Bartlett et al., 2006).*
- *$\psi(u)^{-1} := \ell(0) - (\ell(u) - u\ell'(u)) > 0$ for any $u > 0$.*

This standard assumption is satisfied by several loss functions such as the logistic loss. We remark that the first assumption is used to show the well-definedness of the mean-field Langevin dynamics and derive its discretization error, and also to obtain a uniform generalization error bound through the classical contraction argument (Boucheron et al., 2013; Ledoux and Talagrand, 1991). The second and third assumptions are used to show the convergence of classification error of our estimator.

**Logarithmic Sobolev inequality.** Nitanda et al. (2022); Chizat (2022) showed that the convergence of MFLD crucially relies on properties of the *proximal Gibbs distribution* whose density is given by

$$p_\mu(X) \propto \exp \left( -\frac{1}{\lambda} \frac{\delta F(\mu)}{\delta \mu}(X) \right),$$

where $\mu \in \mathcal{P}$. By the smoothness of the loss function (Assumption 1) and the $\tanh$ activation, we can show that the objective $\mathcal{L}$ has a unique solution $\mu^*$ which is also a proximal Gibbs measure of itself.

**Proposition 1** (Proposition 2.5 of Hu et al. (2019)). *The functional $\mathcal{L}$ has a unique minimizer in $\mathcal{P}_2$ that is absolutely continuous with respect to the Lebesgue measure. Moreover, $\mu^* \in \mathcal{P}_2$ is the optimal solution if and only if $\mu^*$ is absolutely continuous and its density function is given by $p_{\mu^*}$.*

The next question is how fast the solution $\mu_t$ converges to the optimal solution $\mu^*$. As we will see, the convergence of MFLD heavily depends on a *logarithmic Sobolev inequality* (LSI) on $p_\mu$.

---

[3]This should be interpreted in a weak sense, that is, for any continuously differentiable function $\phi$ with a compact support, $\int \phi \mathrm{d}\mu_t - \int \phi \mathrm{d}\mu_s = - \int_s^t \int \nabla \phi \cdot (\nabla \log(\mu_t) - \nabla \frac{\delta F(\mu_t)}{\delta \nu}) \mathrm{d}\mu_\tau \mathrm{d}\tau$.

**Definition 2** (Logarithmic Sobolev inequality). *Let $\mu$ be a probability measure on $(\mathbb{R}^d, \mathcal{B}(\mathbb{R}^d))$. $\mu$ satisfies the LSI with constant $\alpha > 0$ if for any smooth function $\phi : \mathbb{R}^d \to \mathbb{R}$ with $\mathbb{E}_\mu[\phi^2] < \infty$,*

$$\mathbb{E}_\mu[\phi^2 \log(\phi^2)] - \mathbb{E}_\mu[\phi^2] \log(\mathbb{E}_\mu[\phi^2]) \leq \frac{2}{\alpha} \mathbb{E}_\mu[\|\nabla\phi\|_2^2].$$

This is equivalent to the condition that the KL divergence from $\mu$ is bounded by the Fisher divergence: $\int \log(\mathrm{d}\nu/\mathrm{d}\mu)\mathrm{d}\nu \leq \frac{2}{\alpha} \int \|\nabla \log(\mathrm{d}\nu/\mathrm{d}\mu)\|^2 \mathrm{d}\mu$, for any $\nu \in \mathcal{P}$ which is absolutely continuous with respect to $\mu$. The LSI of proximal Gibbs measure can be established via standard perturbation criteria. For $L(\mu)$ with bounded first-variation, we may apply the classical Bakry-Emery and Holley-Stroock arguments (Bakry and Émery, 1985; Holley and Stroock, 1987) (Corollary 5.7.2 and 5.1.7 of Bakry et al. (2014)): If $\|\frac{\delta L(\mu)}{\delta\mu}\|_\infty \leq B$ is satisfied for any $\mu \in \mathcal{P}_2$, then $\mu^*$ and $p_{\mathscr{X}}$ satisfy the LSI with

$$\alpha \geq \lambda_1 \exp\left(-4B/\lambda\right). \tag{5}$$

With the LSI condition on the proximal Gibbs distribution, it is known that the MFLD converges to the optimal solution in an exponential order by using a so-called *Entropy sandwich* technique.

**Proposition 2** (Entropy sandwich (Nitanda et al., 2022; Chizat, 2022)). *Suppose that $\mu_0$ satisfies $\mathcal{L}(\mu_0) < \infty$ and the proximal Gibbs measure $p_{\mu_t}$ corresponding to the solution $\mu_t$ has the LSI constant $\alpha$ for all $t \geq 0$, then the solution $\mu_t$ of MFLD satisfies*

$$\lambda \mathrm{KL}(\mu^*, \mu_t) \leq \mathcal{L}(\mu_t) - \mathcal{L}(\mu^*) \leq \exp(-2\alpha\lambda t)(\mathcal{L}(\mu_0) - \mathcal{L}(\mu^*)),$$

*where $\mu^* = \mathrm{argmin}_{\mu \in \mathcal{P}} \mathcal{L}(\mu)$ (the existence and uniqueness of $\mu^*$ is guaranteed by Proposition 1).*

Hence we know that time horizon $T = O(\frac{1}{\lambda\alpha} \log(1/\tilde{\epsilon}))$ is sufficient to achieve $\tilde{\epsilon} > 0$ accuracy.

**Convergence of the discretized algorithm.** While Proposition 2 only established the convergence rate of the continuous dynamics, similar guarantee can be shown for the discretized setting. Let

$$\mathcal{L}^N(\mu^{(N)}) = N\mathbb{E}_{\mathscr{X} \sim \mu^{(N)}}[F(\mu_{\mathscr{X}})] + \lambda \mathrm{Ent}(\mu^{(N)}),$$

where $\mu^{(N)}$ is a distribution of $N$ particles $\mathscr{X} = (X^i)_{i=1}^N \subset \mathbb{R}^{\bar{d}}$. Let $\mu_\tau^{(N)}$ be the distribution of the particles $\mathscr{X}_\tau = (X_\tau^i)_{i=1}^N$ at the $\tau$-th iteration. Suzuki et al. (2023b) showed that, if $\lambda\alpha\eta \leq 1/4$ and $\eta \leq 1/4$, then for $\bar{B}^2 := \mathbb{E}[\|X_0^i\|^2] + \frac{1}{\lambda\lambda_1}\left[\left(\frac{1}{4} + \frac{1}{\lambda\lambda_1}\right)\bar{R}^2 + \lambda d\right] = O(d + \lambda^{-2})$ and $\delta_\eta := C_1\bar{L}^2(\eta^2 + \lambda\eta)$, where $\bar{L} = 2\bar{R} + \lambda\lambda_1 = O(1)$ and $C_1 = 8(\bar{R}^2 + \lambda\lambda_1\bar{B}^2 + d) = O(d + \lambda^{-1})$,

$$\frac{1}{N}\mathbb{E}[\mathcal{L}^N(\mu_\tau^{(N)})] - \mathcal{L}(\mu^*) \leq \exp\left(-\frac{\lambda\alpha\eta\tau}{2}\right)\left(\frac{\mathbb{E}[\mathcal{L}^N(\mu_0^{(N)})]}{N} - \mathcal{L}(\mu^*)\right) + \frac{4}{\lambda\alpha}\bar{L}^2 C_1(\lambda\eta + \eta^2) + \frac{4C_\lambda}{\lambda\alpha N},$$

where $C_\lambda$ is a constant depending on $\lambda$. In particular, for a given $\tilde{\epsilon} > 0$, the right hand side can be bounded by $\tilde{\epsilon} + \frac{4C_\lambda}{\lambda\alpha N}$ after $T = O\left(\frac{\bar{L}^2 C_1}{\alpha\tilde{\epsilon}} + \frac{\bar{L}\sqrt{C_1}}{\sqrt{\lambda\alpha\tilde{\epsilon}}}\right)\frac{1}{\lambda\alpha} \log(1/\tilde{\epsilon})$ iterations with the step size $\eta = O\left(\left(\frac{\bar{L}^2 C_1}{\alpha\tilde{\epsilon}} + \frac{\bar{L}\sqrt{C_1}}{\sqrt{\lambda\alpha\tilde{\epsilon}}}\right)^{-1}\right)$. Furthermore, the convergence of the loss function can be connected to the convergence of the function value of the neural network as follows,

$$\mathbb{E}_{\mathscr{X}_\tau \sim \mu_k^{(N)}}\left[\sup_{z \in \mathrm{supp}(P_Z)} (f_{\mu_{\mathscr{X}_\tau}}(z) - f_{\mu^*}(z))^2\right]$$

$$\leq \frac{4\bar{L}^2}{\lambda\alpha}\left(\frac{\mathcal{L}^N(\mu_\tau^{(N)})}{N} - \mathcal{L}(\mu^*)\right) + 2\mathbb{E}_{\mathscr{X}_* \sim (\mu^*)^{\otimes N}}\left[\sup_{z \in \mathrm{supp}(P_Z)}\left(\frac{1}{N}\sum_{i=1}^N h_{X_*^i}(z) - \int h_x(z)\mathrm{d}\mu^*(z)\right)^2\right].$$

Here the second term in the right hand side can be bounded by $\frac{32\bar{R}^2}{N}\left[1 + 2\left(\frac{2\bar{R}^2}{(\lambda\lambda_1)^2} + \frac{\bar{d}}{\lambda_1}\right)\right]$ via Lemma 2, if $\|z\| \leq 1$ for any $z \in \mathrm{supp}(P_Z)$ as in the $k$-sparse parity problem. Hence, by taking the number of particles as $N = \epsilon^{-2}[(\lambda\alpha)^{-2} + (\lambda\lambda_1)^{-2} + d/\lambda_1]$ and letting $\tilde{\epsilon} = \lambda\alpha\epsilon^2$ with the choice of $T$ and $\eta$ as described above, we have $\sup_{z \in \mathrm{supp}(P_Z)} |f_{\mu_{\mathscr{X}_T}}(z) - f_{\mu^*}(z)| = O_p(\epsilon)$.

**Assumptions on the model specification.** We restrict ourselves to the situation where a perfect classifier with margin $c_0$ is included in our model class, stated in the following assumption.

**Assumption 2.** *There exists $c_0 > 0$ and $R > 0$ such that the following conditions are satisfied:*

- *For some $\bar{R}$, there exists $\mu^* \in \mathcal{P}$ such that $\mathrm{KL}(\nu, \mu^*) \leq R$ and $L(\mu^*) \leq \ell(0) - c_0$.*

- *For any $\lambda < c_0/R$, the regularized expected risk minimizer $\mu_{[\lambda]} := \mathrm{argmin}\, L(\mu) + \lambda \mathrm{KL}(\nu, \mu)$ satisfies $Y f_{\mu_{[\lambda]}}(X) \geq c_0$ almost surely.*

Importantly, we can apply the same general analysis for different classification problems, as long as Assumption 2 is verified. The advantage of this generality is that we do not need to tailor our convergence proof for individual learning problems. Note that the convergence rate of MFLD is strongly affected by the values of $\bar{R}$ and $R$; therefore, it is crucial to establish this condition using the smallest possible values of $\bar{R}$ and $R$, in order to obtain a tight bound of the classification error. As an illustrative example, we now show that the $k$-sparse parity estimation satisfies the above assumption.

**Example: $k$-sparse parity estimation.** In the $k$-sparse parity setting (Example 1), Assumption 2 is satisfied with constants specified in the following propositions. The proofs are given in Appendix A.

**Proposition 3** ($k$-sparse parity). *Under Assumption 1 and for $\bar{R} = k$, there exists $\mu^* \in \mathcal{P}$ such that*

$$\mathrm{KL}(\nu, \mu^*) \leq c_1 k \log(k)^2 d \ (= R),$$

*and $L(\mu^*) \leq \ell(0) - c_2$, where $c_1, c_2 > 0$ are absolute constants.*

**Proposition 4.** *Under Assumption 1 and the settings of $R$ and $\bar{R}$ given in Proposition 3, if $\lambda < c_2/(2R)$, then $\mu_{[\lambda]}$ satisfies*

$$\max\{\bar{L}(\mu_{[\lambda]}), L(\mu_{[\lambda]})\} \leq \ell(0) - c_2 + \lambda R < \ell(0) - \tfrac{c_2}{2},$$

*and $f_{\mu_{[\lambda]}}$ is a perfect classifier with margin $c_2$, i.e., $Y f_{\mu_{[\lambda]}}(X) \geq \frac{c_2}{2}$.*

In other words, Assumption 2 is achieved with $R = O(k \log(k)^2 d)$, $\bar{R} = k$ and $c_0 = c_2/2$. By substituting these values of $R$ and $\bar{R}$ to our general results presented below, we can easily derive a bound for the classification error of the MFLD estimator.

## 4 Main Result: Annealing Procedure and Classification Error Bound

### 4.1 Annealing Procedure

The convergence rate of MFLD is heavily dependent on the LSI constant which may impose a large computational cost. We alleviate this dependency by employing a novel annealing scheme where we gradually decrease the regularization parameter $\lambda$. In particular, at the $\kappa$-th round, we run the MFLD until (near) convergence with a regularization parameter $\lambda^{(\kappa)} = 2^{-\kappa} \lambda^{(0)}$: $\mu_0 = \mu^{(\kappa-1)}$,

$$\mathrm{d}X_t = -\left[\nabla \tfrac{\delta L(\mu_t)}{\delta \mu}(X_t)\mathrm{d}t + 2\lambda^{(\kappa)}\lambda_1 X_t\right] + \sqrt{2\lambda^{(\kappa)}}\mathrm{d}W_t, \tag{6}$$

which corresponds to minimizing $\mathcal{L}^{(\kappa)}(\mu) := L(\mu) + \lambda^{(\kappa)}\mathrm{KL}(\nu, \mu)$. Then, we obtain a near optimal solution $\mu^{(\kappa)}$ as $\mathcal{L}^{(\kappa)}(\mu^{(\kappa)}) \leq \min_{\mu \in \mathcal{P}} \mathcal{L}^{(\kappa)}(\mu) + \epsilon^*$ for a given $\epsilon^* > 0$. We terminate the procedure after $K$ rounds and obtain $\mu^{(K)}$ as the output.

Suppose that there exists $\mu^*$ such that $\mathrm{KL}(\nu, \mu^*) \leq R$ and $L(\mu^*) \leq \delta^*$. Then, as long as $\lambda^{(\kappa)} \geq \delta^*$ and $\epsilon^* < \delta^*$, we have that

$$L(\mu^{(\kappa)}) + \lambda^{(\kappa)}\mathrm{KL}(\nu, \mu^{(\kappa)}) \leq L(\mu^*) + \lambda^{(\kappa)}\mathrm{KL}(\nu, \mu^*) + \epsilon^* \leq 2\delta^* + \lambda^{(\kappa)}R \leq (R+2)\lambda^{(\kappa)}.$$

Since $\mathcal{L}^{(\kappa)}(\mu_t)$ is monotonically decreasing during the optimization, we always have $L(\mu_t) \leq \mathcal{L}^{(\kappa)}(\mu^{(\kappa-1)}) \leq \mathcal{L}^{(\kappa-1)}(\mu^{(\kappa-1)}) \leq (R+2)\lambda^{(\kappa-1)}$. Now we utilize the following structure on common classification losses to ensure that $\|\delta L(\mu)/\delta \mu\|_\infty$ is small when the loss $L(\mu)$ is small.

**Assumption 3.** *There exists $c_{\mathrm{L}} > 0$, such that, for any $\mu \in \mathcal{P}$, it holds that $\|\frac{\delta L(\mu)}{\delta \mu}\|_\infty \leq c_{\mathrm{L}} \bar{R} L(\mu)$.*

For example, the logistic loss satisfies this assumption:

$$|\partial_u \log(1 + \exp(-u))| = \tfrac{\exp(-u)}{1+\exp(-u)} \leq \log(1 + \exp(-u)).$$

Hence, the condition holds for $c_{\mathrm{L}} = 1$ because $\|\frac{\delta L(\mu)}{\delta \mu}\|_\infty \le \frac{1}{n}\sum_{i=1}^n |\ell'(y_i, f_\mu(z_i))| \|h_{z_i}(\cdot)\|_\infty \le \frac{1}{n}\sum_{i=1}^n \bar{R}\ell(y_i, f_\mu(z_i)) = \bar{R}L(\mu)$. Under this assumption, the Holley–Strook argument yields that the log-Sobolev constant during the optimization can be bounded as

$$\alpha \ge \lambda_1 \exp\left(-\tfrac{4c_{\mathrm{L}}\bar{R}(R+2)\lambda^{(\kappa-1)}}{\lambda^{(\kappa)}}\right) \ge \lambda_1 \exp\left(-8c_{\mathrm{L}}\bar{R}(R+2)\right),$$

which is independent of $\lambda^{(\kappa)}$ and the final accuracy $\delta^*$. Hence, after $K = \log_2(R/(\lambda^{(0)}\epsilon^*))$ round, we achieve that $L(\mu^{(K)}) \le \delta^* + 2\epsilon^*$ where each round takes $T_\kappa = \log(1/\epsilon^*)/(2\alpha\lambda) = O(\log(1/\epsilon^*)\exp(8c_{\mathrm{L}}\bar{R}(R+2))/(\lambda_1\lambda^{(k)}))$ for the continuous time setting.

For the discrete setting, $T_\kappa = O\left(\left(\frac{C_1}{\alpha\epsilon^*} + \frac{\sqrt{C_1}}{\sqrt{\lambda^{(\kappa)}\alpha\epsilon^*}}\right)\log(1/\epsilon^*)\exp(8c_{\mathrm{L}}\bar{R}(R+2))/(\lambda_1\lambda^{(\kappa)})\right)$ iterations in each round is sufficient (where $C_1$ is given for $\lambda^{(K)}$), where the step size is $\eta = O\left(C_1^{-1}\alpha\epsilon^* \wedge C_1^{-1/2}\sqrt{\lambda^{(\kappa)}\alpha\epsilon^*}\right)$. As long as $\epsilon^* = O(\lambda^{(\kappa)}/\alpha)$ for all $\kappa$, the total iteration number can be simplified as $O\left(\log(1/\epsilon^*)\exp(16c_{\mathrm{L}}\bar{R}(R+2))C_1/(\lambda^{(K)}\epsilon^*)\right)$, and the width $N$ (number of particles) can be taken as $N = O([(\lambda^{(K)}\alpha)^{-2} + (\lambda^{(K)})^{-2} + d]/(\epsilon^*)^2) = O\left([\exp(16c_{\mathrm{L}}\bar{R}(R+2))/(\lambda^{(K)})^2 + d]/(\epsilon^*)^2\right)$.

**Remark 1.** *We make the following remarks on the annealing procedure.*

- *The main advantage of this annealing approach is that the exponential factor induced by the LSI constant $\alpha$ is not dependent on the choice of the regularization parameter $\lambda^{(k)}$ (note that the LSI is solely determined by the intermediate solution $\mu_t$). In contrast, the naive Holley–Stroock argument of Eq. (5) imposes exponential dependency on the regularization parameter.*

- *Our annealed algorithm differs from the procedure considered in Chizat (2022, Section 4); importantly, we make use of the structure of classification loss functions to obtain a refined computational complexity analysis.*

## 4.2 Generalization Error Analysis

We utilize the *local Rademacher complexity* (Mendelson, 2002; Bartlett et al., 2005; Koltchinskii, 2006; Giné and Koltchinskii, 2006) to obtain a faster generalization error rate. For the function class of mean-field neural networks, we introduce $\mathcal{F} := \{f_\mu \mid \mu \in \mathcal{P}\}$, and the KL-constrained model class $\mathcal{F}_M(\mu^\circ) := \{f_\mu \mid \mu \in \mathcal{P}, \mathrm{KL}(\mu^\circ, \mu) \le M\}$ for $\mu^\circ \in \mathcal{P}$ and $M > 0$. The Rademacher complexity of a function class $\tilde{\mathcal{F}}$ is defined as

$$\mathrm{Rad}(\tilde{\mathcal{F}}) := \mathbb{E}_{\varepsilon_i, z_i}\left[\sup_{f \in \tilde{\mathcal{F}}} \tfrac{1}{n}\sum_{i=1}^n \epsilon_i f(z_i)\right],$$

where $(z_i)_{i=1}^n$ are i.i.d. observations from $P_Z$ and $(\varepsilon_i)_{i=1}^n$ is an i.i.d. Rademacher sequence ($P(\varepsilon_i = 1) = P(\varepsilon_i = -1) = 1/2)$). We have the following bound on the Rademacher complexity of the function class $\mathcal{F}_M(\mu^\circ)$.

**Lemma 1** (Local Rademacher complexity of $\mathcal{F}_M(\mu^\circ)$, Chen et al. (2020) adapted)**.** *For any fixed $\mu^\circ \in \mathcal{P}$ and $M > 0$, it holds that $\mathrm{Rad}(\mathcal{F}_M(\mu^\circ)) \le 2\bar{R}\sqrt{\frac{M}{n}}$.*

The proof is given in Appendix B.2 in the supplementary material. Combining this local Rademacher complexity bound with the *peeling device* argument (van de Geer, 2000), we can roughly obtain the following estaimte (note that this is an informal derivation):

$$\underbrace{\bar{L}(\hat{\mu}) - \bar{L}(\mu^*) - (\hat{\mu} - \mu^*)\frac{\delta\bar{L}(\mu^*)}{\delta\mu}}_{\text{(II)}} + \underbrace{\lambda\mathrm{KL}(\mu^*, \hat{\mu})}_{\text{(I)}} \lesssim \sqrt{\frac{\mathrm{KL}(\mu^*, \hat{\mu})}{n}} \lesssim \frac{1}{n\lambda} + \lambda\mathrm{KL}(\mu^*, \hat{\mu}), \quad (7)$$

with high probability, where $\hat{\mu} = \mathrm{argmin}_{\mu \in \mathcal{P}} \mathcal{L}(\mu)$, $\mu^* = \mathrm{argmin}_{\mu \in \mathcal{P}} \bar{L}(\mu) + \lambda\mathrm{KL}(\nu, \mu)$, $R$ and $\bar{R}$ are regarded as constants, and the last inequality is by the AM-GM relation. Observe that on the left hand side, we have two non-negative terms (I) and (II). Corresponding to each term, we obtain different types of classification error bounds (Type I and Type II in the following subsection, respectively). Note that there appears a $O(1/n)$ factor in the right hand side, which cannot be

obtained by a vanilla Rademacher complexity evaluation because it only yields an $O(1/\sqrt{n})$ bound. In other words, localization is essential to obtain our fast convergence rate.

We remark that a local Rademacher complexity technique is also utilized by Telgarsky (2023) to derive a $O(1/n)$ rate. They adapted a technique developed for a smooth loss function by Srebro et al. (2010), which requires the training loss $L(\hat{\mu})$ to be sufficiently small, that is, of order $L(\hat{\mu}) = O(1/n)$. In our setting, to achieve such a small training loss, we need to take large $\bar{R}$ such as $\bar{R} = \Omega(\log(n))$. Unfortunately, such a large $\bar{R}$ induces exponentially small log-Sobolev constant like $\alpha \lesssim \exp(-cd\log(n)) = n^{-cd}$. In contrast, our analysis focuses on the local Rademacher complexity around $\mu^*$, and hence we do not require the training loss to be close to 0; instead, it suffices to have a training loss that is close to or smaller than that of $\mu^*$.

### 4.2.1 Type I: Perfect Classification with Exponentially Decaying Error

In the regime of $n = \Omega(1/\lambda^{(K)2})$, we can prove that the MFLD estimator attains a perfect classification with an exponentially converging probability by evaluating the term (I). From Eq. (7) we can establish $\mathrm{KL}(\mu^*, \hat{\mu}) \le O_p(1/(n\lambda^{(K)2}))$; this KL divergence bound can be used to control the $L^\infty$-norm between $f_{\hat{\mu}}$ and $f_{\mu^*}$. Indeed, we can show that $\|f_{\hat{\mu}} - f_{\mu^*}\|_\infty^2 \le 2\bar{R}^2 \mathrm{KL}(\mu^*, \hat{\mu})$ (see the proof of Theorem 1). Then, under the margin assumption of $f_{\mu^*}$ (Assumption 2), we have that $f_{\hat{\mu}}$ also yields a Bayes optimal classifier. More precisely, we have the following theorem.

**Theorem 1.** *Suppose Assumptions 1 and 2 hold. Let $M_0 = (\epsilon^* + 2(\bar{R} + 1))/\lambda^{(K)}$. Moreover, suppose that $\lambda^{(K)} < c_0/R$ and*

$$Q := c_0^2 - \frac{4\bar{R}^2}{n\lambda^{(K)2}}\left[\lambda^{(K)}\left(4\bar{R} + \frac{\lambda^{(K)}}{32\bar{R}^2 n}\right) + 8\bar{R}^2(4 + \log\log_2(8n^2 M_0\bar{R})) + n\lambda^{(K)}\epsilon^*\right] > 0,$$

*then $f_{\hat{\mu}}$ yields perfect classification, i.e., $P(Yf_{\hat{\mu}}(Z) > 0) = 1$, with probability $1 - \exp(-\frac{n\lambda^{(K)2}}{32\bar{R}^4}Q)$.*

The proof is given in Appendix C.1 in the supplementary material. From Proposition 3 we see that the requirement $n \asymp 1/\lambda^{(K)2}$ implies that in the $n \asymp d^2$ regime, Theorem 1 gives a perfect classification guarantee with a failure probability decaying exponentially fast.

### 4.2.2 Type II: Polynomial Order Classification Error

Next we evaluate the classification error bound from term (II) in Eq. (7). In this case, we do not require an $L^\infty$-norm bound as in the Type I analysis above; this results in a milder dependency on $\lambda^{(K)}$ and hence a better sample complexity.

**Theorem 2.** *Suppose Assumptions 1 and 2 hold. Let $\lambda^{(K)} < c_0/R$ and $M_0 = (\epsilon^* + 2(\bar{R}+1))/\lambda^{(K)}$. Then, with probability $1 - \exp(-t)$, the classification error of $f_{\mu^{(K)}}$ is bounded as*

$$P(Yf_{\mu^{(K)}}(Z) \le 0) \le 2\psi(c_0)\left[\frac{8\bar{R}^2}{n\lambda^{(K)}}\left(4 + t + \log\log_2(8n^2 M_0\bar{R})\right) + \frac{1}{n}\left(4\bar{R} + \frac{\lambda^{(K)}}{32\bar{R}^2 n}\right) + \epsilon^*\right].$$

The proof is given in Appendix C.2 in the supplementary material. We notice that the right hand side scales with $O(1/(n\lambda^{(K)}))$, which is better than $O(1/(n\lambda^{(K)2}))$ in Theorem 1; this implies that a sample size linear in the dimensionality is sufficient to achieve small classification error. The reason for such improvement in the $\lambda^K$-dependence is that the stronger $L^\infty$-norm convergence is not used in the proof; instead, only the convergence of the loss is utilized. On the other hand, this analysis does not guarantee a perfect classification.

### 4.2.3 Computational Complexity of MFLD

From the general result in Section 4.1, we can evaluate the computational complexity to achieve the statistical bounds derived above. In both cases (Theorems 1 and 2), we may set the optimization error $\epsilon^* = O(1/(n\lambda^{(K)}))$. Then, the total number of iteration can be

$$\sum_{\kappa=1}^{K} T_\kappa \le O\left((d + \lambda^{(K)-1})n\exp(16c_{\mathrm{L}}\bar{R}(R + 2))\log(n\lambda^{(K)})\right).$$

The width $N$ (the number of particles) can be taken as $N = O((\epsilon^*\lambda^{(K)}\alpha)^{-2}) = O(n^2\alpha^{-2}) = O\left(n^2\exp(16c_{\mathrm{L}}\bar{R}(R + 2))\right)$.

**Corollary 1** (*k*-sparse parity setting). *In the k-sparse parity setting, we may take* $R = O(k\log(k)^2 d)$, $\bar{R} = k$ *and* $\lambda^{(K)} = O(1/R) = O(1/(k\log(k)^2 d))$. *Therefore, the classification error is bounded by*

$$P(Yf_{\mu^{(K)}} < 0) \leq O\left(\frac{k^2 \log(k)^2 d}{n}(\log(1/\delta) + \log\log(n))\right),$$

*with probability* $1 - \delta$. *Moreover, if* $n = \Omega(k^6 \log(k)^4 d^2)$, *then* $P(Yf_{\mu^{(K)}} > 0) = 1$ *with probability*

$$1 - \exp(-\Omega(nk^6 \log(k)^4/d^2)).$$

*As for the computational complexity, we require* $O(k\log(k)^2 dn\log(nd)\exp[O(k^2\log(k)^2 d)])$ *iterations, and the number of particles is* $O(\exp(O(k^2\log(k)^2 d)))$.

**Comparison with prior results.**  In the 2-sparse parity setting, neural network in the kernel (NTK) regime achieves only $O(d^2/n)$ convergence in the classification error, as shown in Ji and Telgarsky (2019) and Telgarsky (2023, Theorem 2.3), whereas we demonstrate that mean-field neural network can improve the rate to $O(d/n)$ via feature learning. Telgarsky (2023) also analyzed the learning of 2-sparse parity problem beyond the kernel regime, and showed that 2-layer ReLU neural network can achieve the best known classification error $O(d/n)$. However, their analysis considered a low-rotation dynamics and assumed convergence at $t \to \infty$, whereas our framework also provides a concrete estimate of the computational complexity. Indeed, the number of iterations can be bounded as $O(dn\log(nd)\exp[O(d)])$. In addition, while we still require exponential width $N = O(n^2 \exp(O(d)))$, such a condition is an improvement over $N = O(d^d)$ in Telgarsky (2023).

Barak et al. (2022) considered a learning method in which one-step gradient descent is performed for the purpose of feature learning, and then a network with randomly re-initialized bias units is used to fit the data. For the *k*-sparse parity problem, they derived a classification error bound of $O(d^{(k+1)/2}/\sqrt{n})$. In contrast, our analysis yields a much better statistical complexity of $O(k^2 \log(k)^2 d/n \wedge \exp(-\Omega(nk^6 \log(k)^4/d^2)))$, which "decouples" the degree $k$ in the exponent of the dimension dependence.

## 5   Numerical Experiment

We validate our theoretical results by numerical experiment on synthetic data. Specifically, we consider the classification of 2-sparse parity with varying dimensionality $d$ and sample size $n$.

Recall that the samples $\{(z_i, y_i)\}_{i=1}^n$ are independently generated so that $z_i$ follows the uniform distribution on $\{\pm 1/\sqrt{d}\}^d$ and $y_i = d\zeta_{i,1}\zeta_{i,2} \in \{\pm 1\}$ $(z_i = (\zeta_{i,1}, \ldots, \zeta_{i,d}))$. A finite-width approximation of the mean-field neural network $\frac{1}{N}\sum_{j=1}^N h_{x_j}(z)$ is employed with the width $N = 2,000$. For each neuron of the network, all parameters are initialized to independently follow the standard normal distribution (meaning that the network is rotation invariant at initialization) and the scaling parameter $\bar{R}$ is set to 15. We set $d$ to take values $5, 10, \cdots, 150$, and $n = 50, 100, \cdots, 2000$. We trained the network using noisy gradient descent with $\eta = 0.2$, $\lambda_1 = 0.1$, and $\lambda = 0.1/d$ (fixed during the whole training) until $T = 10,000$. The logistic loss is used for the training objective.

Figure 1 shows the average test accuracy over five trials. We make the following observations.

- The red line corresponds to the sample size $n = \Theta(d^2)$, above which we observe that almost-perfect classification is achieved. According to Section 4.2 (Type I), the classification error gets exponentially small with respect to $n/d^2$, which predicts very small classification error above the line of $n = \Theta(d^2)$, which matches our experimental result.
- The boundary of test accuracy above $50\%$ is almost linear, as indicated by the blue line. This matches the theoretical conditions (Type II) to obtain the polynomial order classification error in Section 4.2.

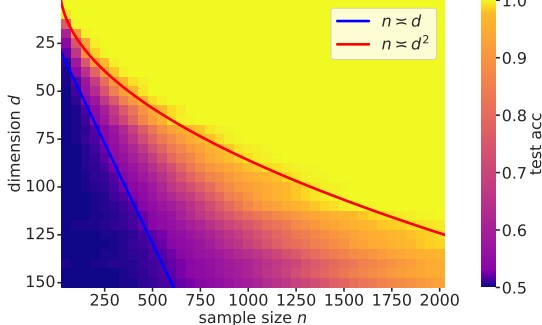

Figure 1: Test accuracy of two-layer neural network optimized by the MFLD to learn a $d$-dimensional 2-sparse parity (XOR) problem.

## 6 Conclusion and Discussion

We provided a general framework to evaluate the classification error of a two-layer neural network trained by the mean-field Langevin dynamics. Thanks to the generality of our framework, an error bound for specific settings can be derived by directly specifying the parameters in Assumption 2 such as $R$, $\bar{R}$, and $c_0$. We also proposed an annealing procedure to alleviate the exponential dependencies in the LSI constant. As a special (but important) example, we investigated the $k$-sparse parity problem, for which we obtained more general and better sample complexity than existing works.

A limitation of our approach is that the required width and number of iterations are exponential with respect to the dimensionality $d$; in contrast, certain tailored algorithms for learning low-dimensional target functions such as Abbe et al. (2023); Chen and Meka (2020) do not require such exponential computation. An important open problem is whether we can reduce the width and number of iterations to $\mathrm{poly}(d)$ for the vanilla noisy gradient descent algorithm. Another interesting direction is to investigate the interplay between structured data and the efficiency of feature learning, as done in Ghorbani et al. (2020); Refinetti et al. (2021); Ba et al. (2023); Mousavi-Hosseini et al. (2023).

## Acknowledgements

The authors thank Matus Telgarsky for discussions and feedback. TS was partially supported by JSPS KAKENHI (20H00576) and JST CREST (JPMJCR2115, JPMJCR2015). KO was partially supported by JST, ACT-X Grant Number JPMJAX23C4, JAPAN. AN was partially supported by JSPS KAKENHI (22H03650).

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

## A Proofs of Propositions 3 and 4

*Proof of Proposition 3.* Remember that

$$h_x(z) = \bar{R}[\tanh(z^\top x_1 + x_2) + 2\tanh(x_3)]/3.$$

Let $b_i = 2i - k$ for $i = 0, \dots, k$, let $\zeta > 0$ be the positive real such that $\mathbb{E}_{u \sim N(0,1)}[2\tanh(\zeta + u)] = 1$ (note that, this also yields $\mathbb{E}_{u \sim N(0,1)}[2\tanh(-\zeta + u)] = -1$ by the symmetric property of $\tanh$ and the Gaussian distribution), and let

$$\xi = [\underbrace{\sqrt{d}, \sqrt{d}, \dots, \sqrt{d}}_{k\text{-dimension}}, \underbrace{0, \dots, 0}_{d-k\text{-dimension}}]^\top \in \mathbb{R}^d.$$

Let

$$\Sigma := \begin{pmatrix} I/(2\lambda_1) & 0 & 0 \\ 0 & 1/(2\lambda_1) & 0 \\ 0 & 0 & 1 \end{pmatrix} \in \mathbb{R}^{(d+1+1)\times(d+1+1)},$$

and $\rho > 1$ be a constant which will be adjusted later on. Then, for $\xi_{2i} := [\log(\rho k)\xi^\top, -\log(\rho k)(b_i - 1), \zeta]^\top \in \mathbb{R}^{\bar{d}}$ and $\xi_{2i+1} := -[\log(\rho k)\xi^\top, -\log(\rho k)(b_i + 1), \zeta]^\top \in \mathbb{R}^{\bar{d}}$ for $i = 0, \dots, k$, we define

$$\hat{\mu}_{2i} := N(\xi_{2i}, \Sigma), \quad \hat{\mu}_{2i+1} := N(\xi_{2i+1}, \Sigma).$$

Then, we can see that, for $z \in \{\pm 1/\sqrt{d}\}^d$, it holds that

$$\mathbb{E}_{x \sim \hat{\mu}_{2i}}[h_x(z)] = \bar{R}\mathbb{E}_{u \sim N(0,1/\lambda_1)}\{\tanh[\log(\rho k)(\langle \xi, z \rangle - (b_i - 1)) + u] + 1\}/3$$

because we have $\langle x_1, z \rangle + x_2 = \log(\rho k)(\langle \xi, z \rangle - (b_i - 1)) + \sum_{j=1}^d u_j z_j + u_{d+1}$ for $x \sim N([\xi^\top, (b_i - 1)]^\top, I/(2\lambda_1))$ where $u_j \sim N(0, 1/(2\lambda_1))$ (i.i.d.) and $\sum_{j=1}^d u_j z_j + u_{d+1}$ obeys the Gaussian distribution with mean 0 and variance $\sum_{j=1}^d \frac{1}{2\lambda_1}\frac{1}{d} + \frac{1}{2\lambda_1} = \frac{1}{\lambda_1}$. In the same vein, we also have

$$\mathbb{E}_{x \sim \hat{\mu}_{2i+1}}[h_x(z)] = -\bar{R}\mathbb{E}_{u \sim N(0,1/\lambda_1)}\{\tanh[\log(\rho k)(\langle \xi, z \rangle - (b_i + 1)) + u] + 1\}/3.$$

Here, define $|z| := |\{i \in \{1, \dots, k\} \mid z_i > 0\}|$ for $z \in \{\pm 1/\sqrt{d}\}^d$ which is the number of positive elements of $z$ in the index set $\{1, \dots, k\}$. For a fixed number $i \in \{0, \dots, k\}$, we let

$$f_1(z; u) = \{\tanh[\log(\rho k)(\langle \xi, z \rangle - (b_i - 1)) + u] + 1\}/3,$$
$$f_2(z; u) = \{\tanh[\log(\rho k)(\langle \xi, z \rangle - (b_i + 1)) + u] + 1\}/3,$$

then we can see that

$$f_1(z; 0) = \begin{cases} O(1/(\rho k)) & (|z| < i), \\ 1 - O(1/(\rho k)) & (|z| \geq i), \end{cases}$$

and

$$f_2(z; 0) = \begin{cases} O(1/(\rho k)) & (|z| < i+1), \\ 1 - O(1/(\rho k)) & (|z| \geq i+1), \end{cases}$$

because $\langle \xi, z \rangle - b_i = \sum_{j=1}^k \text{sign}(z_j)b_i = 2|z| - k - b_i = 2(|z| - i)$. Hence, we have that

$$f(z; u) := f_1(z; u) - f_2(z; u) = \begin{cases} \Omega(1) & (|z| = i), \\ O(1/(\rho k)) & (\text{otherwise}). \end{cases}$$

Then, since $\tanh(u) + 1 = \frac{e^u - e^{-u}}{e^u + e^{-u}} + 1 = \frac{2}{1 + e^{-2u}}$, if $|z| = i$ and $|u| \leq 1/\lambda_1$,

$$f(z; u) \geq \Omega(1),$$

and if $|z| \neq i$ and $|u| \leq \log(\rho k)/2$,

$$f(z; u) \leq O(1/(\rho k)).$$

Therefore, when $|z| = i$,

$$\mathbb{E}_{u \sim N(0, 1/\lambda_1)}[f(z; u)] \geq \int_{-1/\lambda_1}^{1/\lambda_1} f(z; u)g(u)\mathrm{d}u > \Omega(1).$$

where $g$ is the density function of $N(0, 1/\lambda_1)$, and when $|z| \neq i$,

$$\begin{aligned}
\mathbb{E}_{u \sim N(0, 1/\lambda_1)}[f(z; u)] &\leq \int_{-\log(\rho k)/2}^{\log(\rho k)/2} f(z; u)g(u)\mathrm{d}u + \int_{|u| \geq \log(\rho k)/2} f(z; u)g(u)\mathrm{d}z \\
&\leq O(1/(\rho k)) + O\left(\frac{\exp(-\lambda_1 \log(\rho k)^2/2)}{\log(\rho k)}\right) \\
&= O(1/(\rho k)),
\end{aligned}$$

where we used the upper-tail inequality of the Gaussian distribution in the second inequality. Hence, it holds that

$$\hat{f}_i(z) := \mathbb{E}_{x \sim \hat{\mu}_{2i}}[h_x(z)] + \mathbb{E}_{x \sim \hat{\mu}_{2i+1}}[h_x(z)] = \begin{cases} \Omega(k) & (|z| = i), \\ O(1/\rho) & (\text{otherwise}), \end{cases}$$

because $\bar{R} = k$. Therefore, by taking $\rho > 1$ sufficiently large, we also have

$$\hat{f}(z) := \frac{1}{2(k+1)} \sum_{i=0}^{k} (-1)^i \hat{f}_i(z) = \begin{cases} \Omega(1) & (|z| \text{ is even}), \\ -\Omega(1) & (|z| \text{ is odd}), \end{cases}$$

where the constant hidden in $\Omega(\cdot)$ is uniform over any $|z|$. Hence, there exists $c_2' > 0$ such that $Y\hat{f}(Z) > c_2'$ almost surely. Then, if we let $\mu_{\langle a \rangle}(A) := \mu(aA)$ for $a \in \mathbb{R}$, a probability measure $\mu$ and a measurable set $A$, then we can see that $\hat{f}$ is represented as

$$\hat{f}(\cdot) = \mathbb{E}_{x \sim \mu^*}[h_x(\cdot)],$$

where

$$\mu^* = \frac{1}{2(k+1)} \sum_{i=0}^{k} (\hat{\mu}_{2i, \langle (-1)^i \rangle} + \hat{\mu}_{2i+1, \langle (-1)^i \rangle}).$$

Then, by letting $c_2 = \ell(0) - \ell(c_2')$, we have

$$L(\mu^*) \leq \ell(0) - c_2.$$

Next, we bound the KL-divergence between $\nu$ and $\mu^*$. Notice that the convexity of KL-divergence yields that

$$\begin{aligned}
\mathrm{KL}(\nu, \mu^*) &\leq \frac{1}{2(k+1)} \sum_{i=0}^{k} (\mathrm{KL}(\nu, \hat{\mu}_{2i}) + \mathrm{KL}(\nu, \hat{\mu}_{2i+1})) \\
&\leq \lambda_1 \log(\rho k)^2 [\|\xi\|^2 + (\max_i |b_i| + 1)^2] + \log(1/(2\lambda_1)) + \lambda_1(1 + \zeta^2) \\
&= O(\log(k)^2(dk + k^2)) = O(\log(k)^2 dk) \ (= R),
\end{aligned}$$

because $k \leq d$, which gives the assertion. $\qquad\square$

*Proof of Proposition 4.* The first assertion can be seen by $\bar{L}(\mu_{[\lambda]}) \leq \bar{L}(\mu^*) + \lambda\mathrm{KL}(\nu\|\mu^*) \leq \ell(0) - c_2 + \lambda R < \ell(0) - c_2 + c_2/2 = \ell(0) - c_2/2$, by the assumption of $\lambda$ such that $\lambda < c_2/(2R)$. The same argument is also applied to $L(\mu_{[\lambda]})$, that is, $L(\mu_{[\lambda]}) \leq \ell(0) - c_2/2$.

We show that the optimal solution $f_{\mu_{[\lambda]}}$ should satisfy $|f_{\mu_{[\lambda]}}(z)| = |f_{\mu_{[\lambda]}}(z')|$ for all $z, z' \in \{\pm 1/\sqrt{d}\}^d$ by the symmetric property of the distribution. For that purpose, we construct a group action on $\{-1/\sqrt{d}, 1/\sqrt{d}\}^d$. For notational simplicity, we consider $\mathbf{Z}^d := \{-1, +1\}^d$ instead. $\mathbf{Z}^d$ can be equipped with a group structure where the binary operation between $z, z' \in \mathbf{Z}^d$ is given by the element wise product $z \cdot z' = (z_i z_i')_{i=1}^d$. Let $T_j : \mathbf{Z}^d \to \mathbf{Z}^d$ be a group action that flips the $j$-th element $T_j z = (z_1, \ldots, -z_j, \ldots, z_d)$. It is obvious that $T_j$ is bijective. Then, we can show

that there exists a sequence of indices $\sigma(1), \ldots, \sigma(2^d) \in \{1, \ldots, d\}$ such that the "orbit" generated by the chain of the group actions $T_{\sigma(k)} \circ \cdots \circ T_{\sigma(2)} \circ T_{\sigma(1)} z$ for $k = 1, \ldots, 2^d$ covers the entire elements of $\mathbf{Z}^d$ for any initial state $z \in \mathbf{Z}^d$ and the last state turns back to the initial state, that is, $T_{\sigma(2^d)} z = z$. This can be shown by induction. If $d = 1$, we just need to take $\sigma(1) = \sigma(2) = 1$ (indeed, $T_{\sigma(1)} z = -z$ and $T_{\sigma(2)} \circ T_{\sigma(1)} z = z$, which satisfies the condition). Suppose that the argument is true for $d = 1, 2, \ldots, d' - 1$, then we show that it is true for $d = d'$. Indeed, if we write $\sigma'(1), \ldots, \sigma'(2^{d'-1})$ be the corresponding sequence for $d = d' - 1$, then let

$$\sigma(i) = \sigma'(i)$$

for $i = 1, \ldots, 2^{d'-1} - 1$, then the corresponding orbit covers all the elements with the last element is fixed except the initial state. Then, we flip the last coordinate at the $2^{d'-1} + 1$-th step as

$$\sigma(2^{d'-1}) = d,$$

and then we "trace-back" the orbit as

$$\sigma(2^{d'-1} + i) = \sigma'(2^{d'-1} - i),$$

for $i = 1, \ldots, 2^{d'-1} - 1$. By the construction, we notice that, after $2^{d'-1} + 2^{d'-1} - 1$ steps, the state becomes

$$T_{\sigma(2^{d'}-1)} \circ T_{\sigma(2^{d'}-2)} \circ \cdots \circ T_{\sigma(1)} z = [z_1, \ldots, z_{d'-1}, -z_{d'}]^\top.$$

Therefore, by flipping the last coordinate again, the state can come back to the initial state, that is, by setting $\sigma(2^{d'}) = d$, we have $T_{\sigma(2^{d'})} \circ \cdots \circ T_{\sigma(1)} z = z$ while the orbit covers the entire elements of $\mathbf{Z}^{d'}$.

We can define the action $T_j$ to $\{\pm 1/\sqrt{d}\}^d$ in the same manner. We note that the distribution of $Z$ is invariant against the action of $T_j$, that is, $P_Z = T_{j\#} P_Z$ where $T_{j\#}$ is the push-forward induced by $T_j$. Here, let $\hat{\mu} := \mu_{[\lambda]}$ and $\hat{f} = f_{\hat{\mu}}$. We also define an "adjoint operator" $T_j^* : \mathcal{P} \to \mathcal{P}$ for $j \in \{k+1, \ldots, d\}$ such that $f_\mu(T_j z) = f_{T_j^* \mu}(z)$ for all $z \in \{\pm 1/\sqrt{d}\}^d$. We can easily check that there exists $T_j^* \mu$ that satisfies this condition. Indeed, we may take $T_j^* \mu$ such that $T_j^* \mu(A \times B \times C) = \mu(T_j A \times B \times C)$ where $T_j A = \{T_j z \mid z \in A\}$ for any measurable set $A \in \mathbb{B}(\mathbb{R}^d)$, $B \in \mathbb{B}(\mathbb{R})$ and $C \in \mathbb{B}(\mathbb{R})$. Since the probability on product sets uniquely determines the probability on the Borel sigma algebra $\mathbb{B}(\mathbb{R}^d) \times \mathbb{B}(\mathbb{R}) \times \mathbb{B}(\mathbb{R}) = \mathbb{B}(\mathbb{R}^{d+2})$, $T_j^* \mu$ is uniquely determined.

Now, we consider the sequence of the group action $T_{\sigma(1)}, \ldots, T_{\sigma(2^{d-k})}$ constructed above acting on the last $d - k$ coordinates. Since the label is independent of the last $d - k$ coordinates, we have $L(T_{\sigma(1)}^* \circ \cdots \circ T_{\sigma(i)}^* \hat{\mu}) = L(\hat{\mu})$ for all $i = 1, \ldots, 2^{d-k}$. Moreover, the symmetricity of the Gaussian distribution yields that $\mathrm{KL}(\nu, T_j^* \mu) = \mathrm{KL}(\nu, \mu)$. Hence, if we take

$$\hat{\mu}' = \frac{1}{2^{d-k}} \sum_{i=1}^{2^{d-k}} T_{\sigma(1)}^* \circ \cdots \circ T_{\sigma(i)}^* \hat{\mu},$$

then by the convexity of the objective $\mathcal{L}$, we have

$$\mathcal{L}(\hat{\mu}') \leq \frac{1}{2^{d-k}} \sum_{i=1}^{2^{d-k}} \mathcal{L}(T_{\sigma(1)}^* \circ \cdots \circ T_{\sigma(i)}^* \hat{\mu}) \leq \mathcal{L}(\hat{\mu}).$$

Then, by the strong convexity of the objective $\mathcal{L}$ (due to the KL-divergence), we have $\hat{\mu}' = \hat{\mu}$. Then, we notice that

$$f_{\hat{\mu}}(z) = \frac{1}{2^{d-k}} \sum_{i=1}^{2^{d-k}} f_{\hat{\mu}}(T_{\sigma(i)} \circ \cdots \circ T_{\sigma(1)} z),$$

for any $z \in \{\pm 1/\sqrt{d}\}^d$, which means that $f_{\hat{\mu}}(z)$ is constant against any change of the last $d - k$ coordinates.

Next, we consider to change the first $k$ coordinates. We also construct a sequence of the group action $T_{\sigma(1)}, \ldots, T_{\sigma(2^k)}$ acting on the first $k$ coordinates. In this case, the action of $T_j$ on $z$ also flips the

corresponding label. Hence, if $y_{(z)}$ is the label of $z$, then the label for $T_j z$ is $-y_{(z)} = y_{(T_j z)}$. By defining the adjoint operator $T_j^*$ as

$$T_j^* \mu(A \times B \times C) = \mu((-T_j A) \times (-B) \times (-C)),$$

for $j \in \{1, \ldots, k\}$, then by the symmetric shape of $\tanh$, we have

$$y_{(T_j z)} f_{\hat{\mu}}(T_j z) = y_{(z)} f_{T_j^* \hat{\mu}}(z),$$

for any $z \in \{\pm 1/\sqrt{d}\}^d$. We also have that $\mathrm{KL}(\nu, \hat{\mu}) = \mathrm{KL}(\nu, T_j^* \hat{\mu})$ by the symmetry of the Gaussian distribution. Therefore, onece again by the invariance of the $P_Z$ against the action of $T_j$ and the convexity of $\mathcal{L}$, we have that

$$\mathcal{L}(\hat{\mu}') \leq \mathcal{L}(\hat{\mu}),$$

where

$$\hat{\mu}' = \frac{1}{2^k} \sum_{i=1}^{2^k} T_{\sigma(1)}^* \circ \cdots \circ T_{\sigma(i)}^* \hat{\mu}. \tag{8}$$

By the strong convexity of $\mathcal{L}$ due to the KL-divergence term, we have $\mu\mu' = \hat{\mu}$. Hence,

$$y_z f_{\hat{\mu}}(z) = \frac{1}{2^k} \sum_{i=1}^{2^k} y_{(T_{\sigma(i)} \circ \cdots \circ T_1 z)} f_{\hat{\mu}}(T_{\sigma(i)} \circ \cdots \circ T_1 z), \tag{9}$$

for any $z \in \{\pm 1/\sqrt{d}\}^d$, which means that $y_z f_{\hat{\mu}}(z)$ is constant.

Finally, combining Eq. (9) with the fact $\bar{L}(\hat{\mu}) \leq \ell(0) - c_2/2$, we have that $P(Y f_{\hat{\mu}}(Z) > 0) = 1$ and $Y f_{\hat{\mu}}(Z) \geq c_2/2$ almost surely because $\|\ell'\|_\infty \leq 1$. □

## B Proofs of auxiliary lemmas

### B.1 Difference between finite particle network and its integral form for the optimal solution

**Lemma 2.** *Under Assumption 1, if $\|z\| \leq 1$ for any $z \in \mathrm{supp}(P_Z)$, then it holds that*

$$\mathbb{E}_{\mathscr{X}_* \sim (\mu^*)^{\otimes N}} \left[ \sup_{z \in \mathrm{supp}(P_Z)} \left( \frac{1}{N} \sum_{i=1}^N h_{X_*^i}(z) - \int h_x(z) \mathrm{d}\mu^*(z) \right)^2 \right] \leq \frac{16 \bar{R}^2}{N} \left[ 1 + 2 \left( \frac{2 \bar{R}^2}{(\lambda \lambda_1)^2} + \frac{\bar{d}}{\lambda_1} \right) \right].$$

*Proof.* Note that the left hand side can be rewritten as

$$\mathbb{E}_{X_*^i \sim \mu^*} \left[ \left( \sup_{z \in \mathrm{supp}(P_Z)} \left| \frac{1}{N} \sum_{i=1}^N h_{X_*^i}(z) - \int h_x(z) \mathrm{d}\mu^*(z) \right| \right)^2 \right].$$

From the standard argument of concentration inequality corresponding to the Rademacher complexity, it holds that

$$P \left( \sup_{z \in \mathrm{supp}(P_Z)} \left| \frac{1}{N} \sum_{i=1}^N h_{X_*^i}(z) - \int h_x(z) \mathrm{d}\mu^*(z) \right| \right.$$

$$\geq 2 \mathbb{E}_{\epsilon, \mathscr{X}_*} \left[ \sup_{z \in \mathrm{supp}(P_Z)} \left| \frac{1}{N} \sum_{i=1}^N \epsilon_i h_{X_*^i}(z) \right| \right] + \sqrt{\frac{2 t \bar{R}^2}{N}} \right)$$

$$\leq \exp(-t),$$

for any $t > 0$, where $\epsilon = (\epsilon_i)_{i=1}^N$ is an i.i.d. sequence of the Rademacher variable, $\mathscr{X}_* = (X_*^i)_{i=1}^N$ is an i.i.d. sequence generated from $\mu^*$, and the probability is taken with respect to the realization of $\mathscr{X}_*$. Since $\tanh$ is Lipschitz continuous with $|\tanh'| \leq 1$, the contraction inequality of the Rademacher complexity yields that

$$\mathbb{E}_{\epsilon, \mathscr{X}_*} \left[ \sup_{z \in \mathrm{supp}(P_Z)} \left| \frac{1}{N} \sum_{i=1}^N \epsilon_i h_{X_*^i}(z) \right| \right]$$

$$\leq 2\frac{\bar{R}}{3}\mathbb{E}_{\epsilon,\mathscr{X}_*}\left[\sup_{z\in\mathrm{supp}(P_Z)}\left|\frac{1}{N}\sum_{i=1}^N\epsilon_i(z^\top X_{*,1}^i+X_{*,2}^i)\right|\right]+\frac{2\bar{R}}{3}\mathbb{E}_{\epsilon,\mathscr{X}_*}\left[\left|\frac{1}{N}\sum_{i=1}^N\epsilon_i\tanh(X_{*,3}^i)\right|\right]$$

$$\leq 2\frac{\bar{R}}{3}\sqrt{\mathbb{E}_{\epsilon,\mathscr{X}_*}\left[\sup_{z\in\mathrm{supp}(P_Z)}(\|z\|^2+1)\left\|\frac{1}{N}\sum_{i=1}^N\epsilon_i[X_{*,1}^i;X_{*,2}^i]\right\|^2\right]}$$

$$+\frac{2\bar{R}}{3}\sqrt{\mathbb{E}_{\epsilon,\mathscr{X}_*}\left[\left|\frac{1}{N}\sum_{i=1}^N\epsilon_i\tanh(X_{*,3}^i)\right|^2\right]}$$

$$\leq 2\frac{\bar{R}\sqrt{2}}{3}\sqrt{\frac{1}{N}\mathbb{E}[\|X_{*,1}^1\|^2+(X_{*,2}^1)^2]}+\frac{2\bar{R}}{3\sqrt{N}}.$$

To bound the right hand side, we evaluate the moment of $\mu^*$. Since $\mu^*$ is the stationary distribution of the SDE (1). Hence, if $X_0\sim\mu^*$, the process $(X_t)_{t\geq 0}$ obeying the MFLD (1) satisfies $X_t\sim\mu^*$ for any $t\geq 0$. Then, the infinitesimal generator of the MFLD gives that

$$0=\frac{\mathrm{d}}{\mathrm{d}t}\mathbb{E}[\|X_t\|^2]=\mathbb{E}\left[(2X_t)^\top\left(-\nabla\frac{\delta L(\mu_t)}{\delta\mu}(X_t)-\lambda\lambda_1 X_t\right)+2\lambda\bar{d}\right]$$

$$=-2\lambda\lambda_1\mathbb{E}\left[\|X_t\|^2\right]+\mathbb{E}\left[(2X_t)^\top\left(\nabla\frac{\delta L(\mu_t)}{\delta\mu}(X_t)\right)\right]+2\lambda\bar{d}$$

$$=-2\lambda\lambda_1\mathbb{E}\left[\|X_t\|^2\right]+\mathbb{E}\left[\lambda\lambda_1\|X_t\|^2+\frac{1}{\lambda\lambda_1}\left\|\nabla\frac{\delta L(\mu_t)}{\delta\mu}(X_t)\right\|^2\right]+2\lambda\bar{d},$$

which yields that

$$\mathbb{E}\left[\|X_t\|^2\right]\leq\frac{(2\bar{R})^2}{(\lambda\lambda_1)^2}+\frac{2\bar{d}}{\lambda_1}.$$

Therefore,

$$\mathbb{E}_{\epsilon,\mathscr{X}_*}\left[\sup_{z\in\mathrm{supp}(P_Z)}\left|\frac{1}{N}\sum_{i=1}^N\epsilon_i h_{X_*^i}(z)\right|\right]\leq\frac{2\bar{R}}{3\sqrt{N}}\left(2\sqrt{\frac{2\bar{R}^2}{(\lambda\lambda_1)^2}+\frac{\bar{d}}{\lambda_1}}+1\right)=:\frac{\bar{D}}{\sqrt{N}}.$$

Then, for $x\geq 0$, it holds that

$$P\left[\left(\sup_{z\in\mathrm{supp}(P_Z)}\left|\frac{1}{N}\sum_{i=1}^N h_{X_*^i}(z)-\int h_x(z)\mathrm{d}\mu^*(z)\right|\right)^2\geq x\right]$$

$$\leq\min\left\{1,\exp\left(-\frac{N}{4\bar{R}^2}\left(x-\frac{8\bar{D}^2}{N}\right)\right)\right\}.$$

This yields that

$$\mathbb{E}\left[\left(\sup_{z\in\mathrm{supp}(P_Z)}\left|\frac{1}{N}\sum_{i=1}^N h_{X_*^i}(z)-\int h_x(z)\mathrm{d}\mu^*(z)\right|\right)^2\right]$$

$$\leq\int_0^\infty P\left[\left(\sup_{z\in\mathrm{supp}(P_Z)}\left|\frac{1}{N}\sum_{i=1}^N h_{X_*^i}(z)-\int h_x(z)\mathrm{d}\mu^*(z)\right|\right)^2\geq x\right]\mathrm{d}x$$

$$\leq\frac{8\bar{D}^2}{N}+\int_0^\infty\exp\left(-\frac{N}{4\bar{R}^2}\bar{x}\right)\mathrm{d}\bar{x}$$

$$\leq\frac{8\bar{D}^2}{N}+\frac{4\bar{R}^2}{N}\leq\frac{16\bar{R}^2}{N}\left[1+2\left(\frac{2\bar{R}^2}{(\lambda\lambda_1)^2}+\frac{\bar{d}}{\lambda_1}\right)\right].$$

This gives the assertion. $\qquad\square$

## B.2 Proof of Lemma 1

The proof follows the line of Lemma 5.5 of Chen et al. (2020). Let the empirical Rademacher complexity of a function class $\tilde{\mathcal{F}}$ be

$$\widehat{\mathrm{Rad}}(\tilde{\mathcal{F}}) := \mathbb{E}_\epsilon\left[\sup_{\mu \in \tilde{\mathcal{F}}} \frac{1}{n}\sum_{i=1}^n \epsilon_i f(z_i)\right].$$

We can see that $\mathrm{Rad}(\cdot) = \mathbb{E}_{(z_i)_{i=1}^n}[\widehat{\mathrm{Rad}}(\cdot)]$. Then, it holds that

$$\widehat{\mathrm{Rad}}(\mathcal{F}_M(\mu^\circ)) = \frac{1}{\gamma}\mathbb{E}_\epsilon\left[\sup_{\mu \in \mathcal{F}_M(\mu^\circ)} \frac{\gamma}{n}\sum_{i=1}^n \epsilon_i \int h_x(z_i)\mathrm{d}\mu(x)\right]$$

$$\leq \frac{1}{\gamma}\left\{M + \mathbb{E}_\epsilon \log\left[\int \exp\left(\frac{\gamma}{n}\sum_{i=1}^n \epsilon_i h_x(z_i)\right)\mathrm{d}\mu^\circ(x)\right]\right\}$$

$$\leq \frac{1}{\gamma}\left\{M + \log\int \mathbb{E}_\epsilon\left[\exp\left(\frac{\gamma}{n}\sum_{i=1}^n \epsilon_i h_x(z_i)\right)\right]\mathrm{d}\mu^\circ(x)\right\},$$

where the first inequality is by the Donsker-Varadha duality formula of the KL-divergence (Donsker and Varadhan, 1983). The second term in the right hand side can be bounded as

$$\mathbb{E}_\epsilon\left[\exp\left(\frac{\gamma}{n}\sum_{i=1}^n \epsilon_i h_x(z_i)\right)\right] \leq \mathbb{E}_\epsilon\left[\exp\left(\frac{\gamma^2}{n^2}\sum_{i=1}^n \epsilon_i^2 h_x(z_i)^2\right)\right] \leq \exp\left(\frac{\gamma^2}{n}\bar{R}^2\right).$$

Therefore, we have that

$$\widehat{\mathrm{Rad}}(\mathcal{F}_M(\mu^\circ)) \leq \frac{1}{\gamma}\left\{M + \frac{\gamma^2}{n}\bar{R}^2\right\}.$$

The right hand side can be minimized by $\gamma = \sqrt{nM}/\bar{R}$, which yields

$$\widehat{\mathrm{Rad}}(\mathcal{F}_M(\mu^\circ)) \leq 2\bar{R}\sqrt{\frac{M}{n}}.$$

This gives the assersion.

## C  Proofs of Theorems 1 and 2

To show the theorems, we first prepare the following uniform bound, which is crucial to obtain the fast learning rate.

**Lemma 3.** *It holds that*

$$\bar{L}(\mu) - L(\mu) + L(\mu^*) - \bar{L}(\mu^*)$$
$$\leq 2\left[\mathrm{Rad}(\mathcal{F}_{2\mathrm{KL}(\mu,\mu^*)\vee\frac{1}{n}}(\mu^*)) + (2\|f_\mu - f_{\mu^*}\|_\infty \vee n^{-1})\sqrt{\frac{t + \log\log_2(8n^2 M\bar{R})}{n}}\right],$$

*uniformly over $\mu \in \mathcal{P}$ with $\mathrm{KL}(\mu, \mu^*) \leq M$ with probability $1 - \exp(-t)$ for any $t > 0$.*

*Proof.* Let $B_k = M/2^{-k}$ and $C_k = 2\bar{R}/2^{-k'}$ for $k = 0, 1, 2, ...,$ and $\mathcal{F}_{k,k'} := \{\mu \in \mathcal{P} \mid \mathrm{KL}(\mu,\mu^*) \leq, \|f_\mu - f_{\mu^*}\|_\infty \leq C_k\}$. Then, the standard concentration inequality regarding to the Rademacher complexity (Theorem 3.1 of Mohri et al. (2012)) with the contraction inequality (Theorem 11.6 of Boucheron et al. (2013) or Theorem 4.12 of Ledoux and Talagrand (1991)) yields that

$$P\left(\sup_{\mu \in \mathcal{F}_{k,k'}} \bar{L}(\mu) - L(\mu) + L(\mu^*) - \bar{L}(\mu^*) \leq 2\left[\mathrm{Rad}(\mathcal{F}_{B_k}(\mu^*)) + C_k\sqrt{\frac{t}{n}}\right]\right) \geq 1 - \exp(-t),$$

for each $k, k'$. Then, since $\|f_\mu - f_{\mu'}\|_\infty \leq 2\bar{R}$ for any $\mu, \mu' \in \mathcal{P}$, by taking uniform bound for all $k = 0, \ldots, \log_2(n^2)$ and $k' = 0, \ldots, \log_2(2\bar{R}n)$, we have that

$$P\left(\bar{L}(\mu) - L(\mu) + L(\mu^*) - \bar{L}(\mu^*) \leq 2\left[\mathrm{Rad}(\mathcal{F}_{2\mathrm{KL}(\mu,\mu^*)\vee\frac{1}{n}}(\mu^*)) + (2\|f_\mu - f_{\mu^*}\|_\infty \vee \frac{1}{n})\sqrt{\frac{t}{n}}\right],\right.$$

$$\left.\forall f \in \mathcal{F}_n(\mu^*)\right) \geq 1 - (2 + \log_2(nM) + \log_2(2\bar{R}n))\exp(-t).$$

By resetting $t \leftarrow t + \log\log_2(8n^2 M\bar{R})$, we obtain the assertion. $\qquad\square$

### C.1 Proof of Theorem 1

By Assumption 2, we have

$$\max\{\bar{L}(\mu_{[\lambda^{(K)}]}), L(\mu_{[\lambda^{(K)}]})\} \leq \ell(0) - c_0,$$

and

$$P\left(Yf_{\mu_{[\lambda^{(K)}]}}(Z) \geq c_0\right) = 1.$$

Let $\hat{\mu} := \mu^{(K)}$ and $\mu^* := \mu_{[\lambda^{(K)}]}$. By the optimality condition of $\hat{\mu}$, we have

$$L(\hat{\mu}) + \lambda^{(K)}\mathrm{KL}(\nu, \hat{\mu}) \leq L(\mu_{[\lambda^{(K)}]}) + \lambda^{(K)}\mathrm{KL}(\nu, \mu_{[\lambda^{(K)}]}) + \epsilon^* < 1/2 - 2\delta^* = c_0.$$

Moreover,

$$\bar{L}(\hat{\mu}) + \lambda^{(K)}\mathrm{KL}(\nu, \hat{\mu}) - (\bar{L}(\mu^*) + \lambda^{(K)}\mathrm{KL}(\nu, \mu^*))$$
$$\leq L(\hat{\mu}) + \lambda^{(K)}\mathrm{KL}(\nu, \hat{\mu}) - (L(\mu^*) + \lambda^{(K)}\mathrm{KL}(\mu^*, \nu)) + (\bar{L}(\hat{\mu}) - L(\hat{\mu}) + L(\mu^*) - \bar{L}(\mu^*))$$
$$\leq \epsilon^* + (\bar{L}(\hat{\mu}) - L(\hat{\mu}) + L(\mu^*) - \bar{L}(\mu^*)).$$

By the optimality of $\mu^*$, the right hand side can be lower bounded as

$$\bar{L}(\hat{\mu}) + \lambda^{(K)}\mathrm{KL}(\nu, \hat{\mu}) - (\bar{L}(\mu^*) + \lambda^{(K)}\mathrm{KL}(\nu, \mu^*))$$
$$\geq \bar{L}(\hat{\mu}) - \bar{L}(\mu^*) - (\hat{\mu} - \mu^*)\frac{\delta\bar{L}(\mu^*)}{\delta\mu} + \lambda^{(K)}\mathrm{KL}(\mu^*, \hat{\mu}).$$

Then, we have $\mathrm{KL}(\mu^*, \hat{\mu}) \leq (\epsilon^* + 2(\bar{R} + 1))/\lambda^{(K)}$. Hence, the uniform bound on the Rademacher complexity (Lemma 3) with $M_0 = (\epsilon^* + 2(\bar{R} + 1))/\lambda^{(K)}$, it holds that

$$\bar{L}(\hat{\mu}) - L(\hat{\mu}) + L(\mu^*) - \bar{L}(\mu^*)$$
$$\leq 2\left[\mathrm{Rad}(\mathcal{F}_{2\mathrm{KL}(\mu^*,\hat{\mu})\vee\frac{1}{n}}(\mu^*)) + (2\|f_{\hat{\mu}} - f_{\mu^*}\|_\infty \vee n^{-1})\sqrt{\frac{t + \log\log_2(8n^2 M_0\bar{R})}{n}}\right],$$

with probability $1 - \exp(-t)$ for any $t > 0$. Moreover, Lemma 1 gives

$$\mathrm{Rad}(\mathcal{F}_{2\mathrm{KL}(\hat{\mu},\mu^*)\vee\frac{1}{n}}(\mu^*)) \leq 2\bar{R}\sqrt{\frac{2\mathrm{KL}(\mu^*, \hat{\mu}) \vee n^{-1}}{n}}.$$

Therefore, by setting $t = s\sqrt{n}$ with $s < 1$, we obtain that

$$\bar{L}(\hat{\mu}) - \bar{L}(\mu^*) - (\hat{\mu} - \mu^*)\frac{\delta\bar{L}(\mu^*)}{\delta\mu} + \lambda^{(K)}\mathrm{KL}(\mu^*, \hat{\mu})$$
$$\leq 2\left[2\bar{R}\sqrt{\frac{2\mathrm{KL}(\mu^*, \hat{\mu}) \vee n^{-1}}{n}} + (2\|f_{\hat{\mu}} - f_{\mu^*}\|_\infty \vee n^{-1})\sqrt{\frac{s\sqrt{n} + \log\log_2(8n^2 M_0\bar{R})}{n}}\right]$$
$$\leq \frac{1}{4}\lambda^{(K)}\mathrm{KL}(\mu^*, \hat{\mu}) + \frac{32\bar{R}^2}{n\lambda^{(K)}} + \frac{4\bar{R}}{n}$$
$$\quad + \lambda^{(K)}\frac{\|f_{\hat{\mu}} - f_{\mu^*}\|_\infty^2}{8\bar{R}^2} + \frac{\lambda^{(K)}}{32\bar{R}^2 n^2} + \frac{8\bar{R}^2}{\lambda^{(K)}}\frac{s\sqrt{n} + \log\log_2(8n^2 M_0\bar{R})}{n},$$

with probability $1 - \exp(-s\sqrt{n})$, where a arithmetic-geometric mean relation and an inequality $(a \vee b)^2 \leq a^2 + b^2$ $(a, b \geq 0)$ are used in the last inequality. Here, we know that the Hellinger distance between $\mu^*$ and $\hat{\mu}$ can be bounded as

$$2 \int (\sqrt{\mu^*} - \sqrt{\hat{\mu}})^2 \mathrm{d}x = 2\mathrm{d}_{\mathrm{H}}(\mu^*, \hat{\mu}) \leq \mathrm{KL}(\mu^*, \hat{\mu}).$$

Hence,

$$
\begin{aligned}
\|f_{\hat{\mu}} - f_{\mu^*}\|_\infty^2 &= \|\int h_x(\cdot)(\hat{\mu}(x) - \mu^*(x))\mathrm{d}x\|_\infty^2 \\
&= \left\| \int h_x(\cdot)(\sqrt{\hat{\mu}(x)} + \sqrt{\mu^*(x)})(\sqrt{\hat{\mu}(x)} - \sqrt{\mu^*(x)})\mathrm{d}x \right\|_\infty^2 \\
&\leq \left\| \int h_x(\cdot)^2(\sqrt{\hat{\mu}(x)} + \sqrt{\mu^*(x)})^2\mathrm{d}x \right\|_\infty \int (\sqrt{\hat{\mu}(x)} - \sqrt{\mu^*(x)})^2\mathrm{d}x \\
&\leq \left\| 2 \int h_x(\cdot)^2(\hat{\mu}(x) + \mu^*(x))\mathrm{d}x \right\|_\infty \int (\sqrt{\hat{\mu}(x)} - \sqrt{\mu^*(x)})^2\mathrm{d}x \\
&\leq 4\bar{R}^2 \mathrm{d}_{\mathrm{H}}(\mu^*, \hat{\mu}) \leq 2\bar{R}^2 \mathrm{KL}(\mu^*, \hat{\mu}).
\end{aligned}
$$

By summarizing the argument above and noticing $\bar{L}(\hat{\mu}) - \bar{L}(\mu^*) - (\hat{\mu} - \mu^*)\frac{\delta\bar{L}(\mu^*)}{\delta\mu} \geq 0$, it holds that

$$
\begin{aligned}
&\bar{L}(\hat{\mu}) - \bar{L}(\mu^*) - (\hat{\mu} - \mu^*)\frac{\delta\bar{L}(\mu^*)}{\delta\mu} + \frac{1}{2}\lambda^{(K)}\mathrm{KL}(\mu^*, \hat{\mu}) \\
&\leq \frac{1}{2}\lambda^{(K)}\mathrm{KL}(\mu^*, \hat{\mu}) \\
&\leq \frac{32\bar{R}^2}{n\lambda^{(K)}} + \frac{4\bar{R}}{n} + \frac{\lambda^{(K)}}{32\bar{R}^2 n^2} + \frac{8\bar{R}^2}{\lambda^{(K)}}\frac{s\sqrt{n} + \log\log_2(8n^2 M_0 \bar{R})}{n},
\end{aligned}
$$

with probability $1 - \exp(-s\sqrt{n})$. Then, letting $s = \tau\lambda^{(K)}$, then we have

$$
\begin{aligned}
&\|f_{\hat{\mu}} - f_{\mu^*}\|_\infty^2 \\
&\leq 4\bar{R}^2 \left( \frac{32\bar{R}^2}{n\lambda^{(K)2}} + \frac{4\bar{R}}{n\lambda^{(K)}} + \frac{1}{32\bar{R}^2 n^2} + 8\bar{R}^2\frac{\tau\lambda^{(K)}\sqrt{n} + \log\log_2(8n^2 M_0 \bar{R})}{n\lambda^{(K)2}} \right) \\
&\leq \frac{4\bar{R}^2}{n\lambda^{(K)2}} \left( 32\bar{R}^2 + 4\bar{R}\lambda^{(K)} + \frac{\lambda^{(K)2}}{32\bar{R}^2 n} + 8\bar{R}^2[\tau\sqrt{n\lambda^{(K)2}} + \log\log_2(8n^2 M_0 \bar{R})] \right),
\end{aligned}
$$

with probability $1 - \exp(-\tau\lambda^{(K)}\sqrt{n})$ for any $\tau > 0$. Hence, if we take $n$ sufficiently large and $\tau$ sufficiently small so that the right hand side is smaller than $c_0^2$, then we have

$$P\left(Y f_{\hat{\mu}}(Z) > 0\right) = 1.$$

More precisely, if we let

$$s_n = \frac{n\lambda^{(K)2}}{32\bar{R}^4} \left\{ c_0^2 - \frac{4\bar{R}^2}{n\lambda^{(K)2}} \left[ \lambda^{(K)}\left( 4\bar{R} + \frac{\lambda^{(K)}}{32\bar{R}^2 n} \right) + 8\bar{R}^2(4 + \log\log_2(8n^2 M_0 \bar{R})) \right] \right\},$$

and $s_n > 0$, then $P\left(Y f_{\hat{\mu}}(Z) > 0\right) = 1$ with probability $1 - \exp(-s_n)$.

## C.2 Proof of Theorem 2

Let $\hat{\mu} := \mu^{(K)}$ and $\mu^* := \mu_{[\lambda^{(K)}]}$. From the proof of Theorem 1, with probability $1 - \exp(-t)$, it holds that

$$
\begin{aligned}
&\frac{32\bar{R}^2}{n\lambda^{(K)}} + \frac{4\bar{R}}{n} + \frac{\lambda^{(K)}}{32\bar{R}^2 n^2} + \frac{8\bar{R}^2}{n\lambda^{(K)}}\left( t + \log\log_2(8n^2 M_0 \bar{R}) \right) \\
&\geq \bar{L}(\hat{\mu}) - \bar{L}(\mu^*) - (\hat{\mu} - \mu^*)\frac{\delta\bar{L}(\mu^*)}{\delta\mu} + \frac{1}{2}\lambda^{(K)}\mathrm{KL}(\mu^*, \hat{\mu}) \\
&\geq \frac{1}{2}\left[ \bar{L}(\hat{\mu}) - \bar{L}(\mu^*) - (\hat{\mu} - \mu^*)\frac{\delta\bar{L}(\mu^*)}{\delta\mu} + \lambda^{(K)}\mathrm{KL}(\mu^*, \hat{\mu}) \right]
\end{aligned}
$$

$$\geq \frac{1}{2}\left[\bar{L}(\hat{\mu}) - \bar{L}(\mu^*) - (\hat{\mu} - \mu^*)\frac{\delta\bar{L}(\mu^*)}{\delta\mu}\right].$$

Since $Yf_{\mu^*}(X) \geq c_0$ almost surely, the Markov's inequality yields that

$$P(Yf_{\hat{\mu}} < 0)$$
$$\leq \frac{1}{\ell(0) - (-c_0\ell'(c_0) + \ell(c_0))}\left(\bar{L}(\hat{\mu}) - \bar{L}(\mu^*) - (\hat{\mu} - \mu^*)\frac{\delta\bar{L}(\mu^*)}{\delta\mu}\right)$$
$$= 2\psi(c_0)\frac{1}{2}\left(\bar{L}(\hat{\mu}) - \bar{L}(\mu^*) - (\hat{\mu} - \mu^*)\frac{\delta\bar{L}(\mu^*)}{\delta\mu}\right).$$

This yields the assertion.

## D  Additional Experiments

In addition to the 2-sparse parity problem, here we give additional experiments for the $k$-sparse parity problems with $k = 3$ and $k = 4$. The setting is the same with the main text, except that the width, step size, the number of iterations are modified to $N = 2,000$, $\eta = 0.2$, and $T = 10,000$, and we take different grids of $n$ and $d$. We ran the experiment 10 times and plotted the mean for each $n$ and $d$. The results are provided in Figure 2.

For $k = 3$, which is shown in Figure 2 (a), we can see that there appear phase-transition lines of $n = \Theta(d^2)$ and $n = \Theta(d)$, which correspond to the test accuracy of around $90\%$ and the test accuracy of around $70\%$, respectively. The former almost perfect classification means the Type I convergence in 4.2, where the classification error gets exponentially small with respect to $n/d^2$. The latter means the Type II polynomial convergence.

For $k = 4$, presented in Figure 2 (b), only the linear phase-transition, that corresponds to the test accuracy of around $70\%$, is observed.

Together with Section 5, we conclude that the our theoretical results are experimentally verified for the 2-sparse parity problem as well as higher-order sparse parity problems. The reason why the quadratic line was not observed in $k = 4$ would be partially because the hidden coefficients and the required number of steps are exponentially large with respect to $k$.

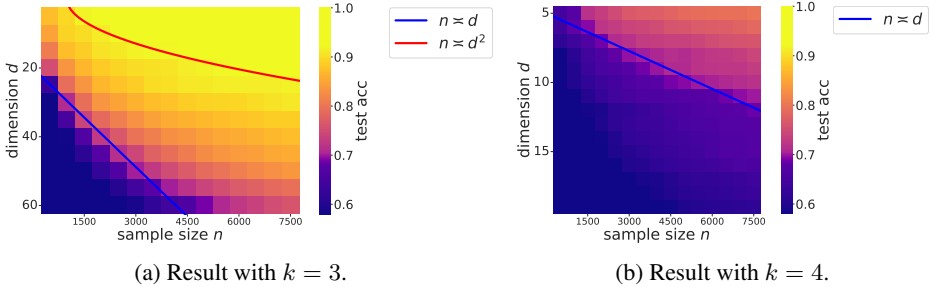

(a) Result with $k = 3$.  (b) Result with $k = 4$.

Figure 2: Test accuracy of two-layer neural network trained by MFLD to learn a $d$-dimensional $k$-sparse parity problems.

