# OpenReview forum: "Feature learning via mean-field Langevin dynamics: classifying sparse parities and beyond"
_NeurIPS.cc/2023/Conference — NeurIPS 2023 poster_

### Official Review · Reviewer_zm9j · 2023-06-29

**Soundness:** 3 good
**Presentation:** 3 good
**Contribution:** 3 good
**Rating:** 6
**Confidence:** 3

**Summary:**

This paper characterizes the generalization of neural networks, and prove efficient sample complexity guarantees in the mean-field region with the presence of feature learning. It presents a general framework to establish sample complexity of MFLD for binary classification problem. Such framework can be used to obtain generalization guarantee for the learned neural network in discrete-time and finite-width settings. When applied to the k-sparse parity problem, the proposed framework yields an improved convergence rate and dimension dependence.


**Strengths:**


The technique used in this analysis is quite interesting and unique. Unlike existing margin bounds for neural networks that rely on norm control, this study takes a different approach by considering that MFLD optimizes the distribution of parameters rather than the parameters themselves. This perspective allows for an improved analysis of the sample complexity and convergence rate.

**Weaknesses:**

1. This study is focus on binary-classification problem. Definitely it could be a good starting point, but at the same time, it is hard to argue that the results for binary-classification problem is very significant.

2. The authors focus on k-sparse parity problem (2-sparse in particular) as an illustrative example throughout the paper. While I understand the 2-sparse parity problem is a well studied problem with a rich body of literature, I still do not quite understand the significance or the motivation of studying this particular problem.

3. The width and number of iterations required in this analysis is of $O(e^d)$ which I think is far from reasonable in practice.


**Questions:**

1. What is the difference of the last two rows in Table 1? Why they have the identical region/method, width, and number of iterations, but different class error?

2. The work done by Telgarsky  (2023) requires width of $O(2^d)$ not $O(d^d)$ right?




**Limitations:**

I believe the authors have a sufficient discussion on the limitations.

A couple of suggestions:

  - $d$ has been used in the introduction several times but never been properly defined in this section (it has been defined in the table 1 though).

  - In Assumption 2, last inequality $L(\mu^*) \leq log(0)-c_0$, I believe $log(0)$ here is a typo. The same typo also appeared in Proposition 3.

---

> ### Author Rebuttal · Authors · 2023-08-09
>
> Thank you very much for your insightful comments.
>
> **Q:** *While I understand the 2-sparse parity problem is a well studied problem with a rich body of literature, I still do not quite understand the significance or the motivation of studying this particular problem.*
> **A:** The parity setting is a canonical example of learning a low-dimensional signal (since $k\ll d$) from high-dimensional data. The low-dimensionality of target function may be exploited by *adaptive* learning procedures (as opposed to fixed bases models such as kernel methods), and thus this function class has been studied in recent works to demonstrate the advantage of representation learning in neural networks (e.g., see Wei et al. 2019).
> In addition, the $k$-sparse parity function corresponds to the Fourier bases on the discrete lattice. Hence, analyzing this problem can give some insight to the understanding of function estimation in the Euclidean space with orthogonal basis expansions such as Hermite polynomial and Fourier basis.
>
> **Q:** *The width and number of iterations required in this analysis is of $O(e^d)$*
> **A:** Indeed this is one main drawback of the current mean-field analysis. This being said, as shown in Table 1, our result already represents a noticeable improvement over existing mean-field analyses, in terms of both the width (prior results required $N=\mathcal{O}(d^d)$) and the iteration complexity (prior works assumed $t\to\infty$), even though our results can handle the more general $k$-parity setting.
> Therefore, we believe that our analysis serves as an important step toward more efficient (quantitative) learning guarantees for mean-field neural networks.
>
> **Q:** *What is the difference of the last two rows in Table 1?*
> **A:** The two rows correspond to different sample size regimes $n \gg d^2$ and $n \gg d$; in other words, they can be put together into one row where the classification error becomes $\min\\{d/n,\exp(-O(\sqrt{n}/d))\\}$.
>
> **Q:** *The work done by Telgarsky (2023) requires width of $O(2^d) $ not $O(d^d)$ right?*
> **A:** The required width in the table is coming from Table 1 of the arXiv version of Telgarsky (2023).
> However, we noticed that it has been modified to $O(d^{d/2})$ in its ICLR camera ready version.
> Accordingly to the paper, it should be $O(d^{d/2})$ instead of $O(2^d)$.
>
> **Q:** *Typos and suggestions.*
> **A:** Thank you very much for your suggestions. We will modify our manuscript according to your suggestions.
>
> We would be happy to clarify any concerns or answer any questions that may come up during the discussion period.

---

> > ### Comment · Reviewer_zm9j · 2023-08-14
> >
> > I would like to thank the authors for their rebuttal. After carefully considering the rebuttal and the other reviews, I have chosen to keep my score.

---

### Official Review · Reviewer_uexs · 2023-07-06

**Soundness:** 3 good
**Presentation:** 3 good
**Contribution:** 3 good
**Rating:** 6
**Confidence:** 4

**Summary:**

This paper considers the problem of learning the k-sparse parity problem with a two-layer network in mean-field regime. The main results are in two folds: 1. the authors proposed an annealing method to obtain a better rate of convergence. 2. the authors compute the classification error via computing the local Rademacher complexity.

**Strengths:**

1. The problem of learning a k-sparse parity function in a mean-field regime seems to be interesting.

2. Proposition 4, which gives a characterization of the margin at the stationary distribution seems to be novel and interesting to me.

3. The annealing methods proposed in this paper seem to be interesting, which gives a way to obtain a better convergence rate via bounding log-Sobolev constant properly.

4. This paper is in general well-written, the main theoretical results are stated clearly, and I feel it is rather easy to understand the main results of this paper.

**Weaknesses:**

I haven't observed an obvious weakness in this paper, but I do have a few questions and comments, please refer to the "Questions" part.

**Questions:**

1. Regarding the annealing methods: If I understand the annealing methods correctly, the intuition is that under assumption 3, the log Sobolev constant is actually controlled by the loss, and since the loss is decreasing along the trajectory, the log Sobolev constant will be larger than the trivial bounds in ( Nitanda et al., 2022; Chizat, 2022). Thus it seems to me that annealing is added for the purpose of controlling the loss properly, rather than fundamentally speeding up the dynamics (in fact, annealing does not improve the convergence rate of the MFLD in general, see e.g. Chizat, 2022 in your reference ). Also, I'm wondering if it is possible to obtain a faster convergence through different annealing methods.?



2. In Proposition 4, you characterize the margin of the stationary distribution, with small enough regularization. I'm wondering if this properties still hold in the limit of $\lambda \rightarrow 0$?

3. Generality of the results:  I'm wondering if it's possible to generalize the result to any bounded smooth activation functions, especially propositions 3 and 4, and Theorem 1 and 2?


Minor points and typos:

1. Line 169: $\log(0) \rightarrow \ell(0)$

2. Line 203: $\lambda^{(k)} \rightarrow \lambda^{(\kappa)}$

3. In the statement of Theorem 1, you have an assumption $Q: = ... > 0$. Given a specific setting (e.g. for logistic loss and taking the parameters in propositions 3 and 4), is this assumption verified?

**Limitations:**

No limitations and potential negative societal impact.

---

> ### Author Rebuttal · Authors · 2023-08-09
>
> Thank you very much for your insightful comments. We address the technical points below.
>
> **Q:** *It seems to me that annealing is added for the purpose of controlling the loss properly, rather than fundamentally speeding up the dynamics (in fact, annealing does not improve the convergence rate of the MFLD in general, see e.g. Chizat, 2022 in your reference). Also, I'm wondering if it is possible to obtain a faster convergence through different annealing methods?*
> **A:** Recall that the log-Sobolev constant is strongly dependent on the value of the empirical loss, and we expect an exponential dependency on the sample size $n$ in the LSI constant without proper annealing.  We note that the regularization coefficient $\lambda$ (a.k.a. temparature) should be properly small to achieve the perfect classification, which exponentially deteriorates the convergence rate without annealing.
> Our main insight is that by controlling the temperature parameter carefully, we can avoid such as exponential dependency.
> This property is special to the logistic loss (its derivative is bounded by its absolute value) and does not hold in general settings.
> In contrast, Chizat (2022) did not take this point into account, and consequently the computational complexity is worse than ours -- we will include a more detailed discussion in the revision.
> There is definitely a possibility that better convergence rate can be obtained by a different annealing method, which we leave as future work.
>
> **Q:** *In Proposition 4, you characterize the margin of the stationary distribution, with small enough regularization. I'm wondering if this properties still hold in the limit of $\lambda \to 0$?*
> **A:** This is a good point. We believe that it holds also for $\lambda=0$.
> This requires minor modification of our proof, because the current analysis relies on the uniqueness of the optimal solution $\mu^*$ which does not hold for $\lambda = 0$. However, this point can be overcome by showing uniqueness of the ``function value'' $f_{\mu^*}(x)$ for optimal solutions instead of the uniqueness of the optimal measure $\mu^*$, which could be shown by the strong convexity of the logistic loss.
>
>
> **Q:** *Generality of the results: I'm wondering if it's possible to generalize the result to any bounded smooth activation functions, especially propositions 3 and 4, and Theorem 1 and 2?*
> **A:** These statements may not be generally true because the activation function should have sufficiently large expressive power.
> This being said, we expect similar results for wide range of activation functions as long as they have symmetricity such as $h_x = -h_{-x}$,
> even though our proof utilizes the explicit form of the $\tanh$ activation.
>
> **Q:** *Minor points and typos*
> **A:** Thank you so much for pointing them out. We will correct them in the revision.
>
> **Q:** *Condition on $Q$*
> **A:** Yes, this can be verified easily. Essentially, this condition asserts that ``the sample size $n$ is sufficiently large'' because the parameters $c_0$ and $\bar{R}$ can be seen as constants.
> More specifically, if $n \gg (\lambda^{(K)})^{-2}$, then the condition holds.
>
> We would be happy to clarify any concerns or answer any questions that may come up during the discussion period.

---

> > ### Comment · Reviewer_uexs · 2023-08-18
> > **Keep my scores as is**
> >
> > I thank the authors for the detailed explanation of my questions and concerns.
> >
> > After reading the rebuttals, I believe this paper is technically solid, and the results are interesting.  The main limitation in my opinion is the generality of the techniques since it seems to me that the proofs rely on the properties of logistic loss and specific activation function. However, since this paper consider a very specific problem ( learning a sparse parity function) using a very specific achitecture, the abovementioned limitations are not issues for me. Besides, the results from this paper is interesting to me.
> >
> > In conlusion, I think this is a technically solid providing interesting results despite a few limitations, thus I will keep my scores as is.

---

### Official Review · Reviewer_1qsa · 2023-07-07

**Soundness:** 2 fair
**Presentation:** 2 fair
**Contribution:** 2 fair
**Rating:** 6
**Confidence:** 2

**Summary:**

The paper conducts a theoretical study of the generalization error and sample complexity of two layer neural network training with Mean Field Langevin Dynamics (MFLD) or (informally a version of )noisy gradient descent. Specifically the paper specializes the generalization error for subset parity problem to derive bounds. Empirical results are provided to illustrate the agreement between theory and empirical observations with synthetic data

**Strengths:**

(Note that I am not very familiar with the details of the subject matter and also related work)

- Analytical generalization bounds may be of interest to the theoretical community especially given the recent interest seen with using subset parity as a dataset in analysis
- The paper does a nice job of comparing existing and their classification error bounds for k-sparse (subset) parity problem to highlight their contribution. The fact that their bound decouples error from k is notable to this reader
- This reviewer appreciates the experimental results in the paper. This figure summarizes the analytical findings included in Section 4


**Weaknesses:**

(Note that I am not very familiar with the details of the subject matter and also related work)

- The paper appears to be spend a lot of real estate with preliminaries at the cost of sacrificing clarity of presentation of the main result. In section 4.1, I do not quite follow how the choice of regularization parameter reduces complexity
- While analysis does decouple the dependence on k in the k-sparse parity task, this appears to be at the cost of the network width and the number of optimization steps which are both exponential compared to the work by Barak et al. (NeurIPS 2022). I am left wondering about the usefulness of this result to the community
- (minor) I glanced at the code for the experiments and noticed that m = 2000. I assume this is the width value used in the paper. The paper states a value of 10_000 so I want to make sure that the shared code has a typo. It would be nice if the code is updated to reflect the exact value used in the paper for clarity

**Questions:**

Please check the weaknesses section

- Please consider updating Section 4 to make new work clear by perhaps moving some preliminaries to the supplement
- Please update code to ensure the results included in the paper can be reproduced without any change(s)

**Limitations:**

The authors do a very nice job of noting weaknesses in their discussion. No more changes necessary

---

> ### Author Rebuttal · Authors · 2023-08-09
>
> Thank you for carefully reading our paper and giving insightful comments.
>
> **Q:** *The paper appears to be spend a lot of real estate with preliminaries.*
> **A:** Thank you for the suggestion. Our goal is to make the main text as self-contained as possible; hence we introduced the basics of the mean-field Langevin dynamics (distribution dependent SDE, log-Sobolev inequality, and so on) before presenting the main result. In the revision, we will improve the presentation and reorganize the main text.
>
> **Q:** *I do not quite follow how the choice of regularization parameter reduces complexity.*
> **A:** As seen in Proposition2, the LSI constant $\alpha$ controls the convergence rate. And this rate could exponentially deteriorate to achieve the perfect classification since the coefficient $\lambda$ appeared in Eq. (5) should be properly small depending on the number of examples $n$. However, noting the exponent of Eq. (5) is $-B/\lambda$ which is essentially the ratio of the loss and $\lambda$, such exponential deterioration could be resolved by the annealing for controlling the term $-B/\lambda$.
>
> **Q:** *Comparison to Barak et al. (NeurIPS 2022).*
> **A:** First, we note that our bound gives much better sample complexity than Barak et al. (2022) -- their bound on the classification error is $O(d^{(k+1)/2}/\sqrt{n})$ while ours is $O(d/n)$. While we achieve such an improved sample complexity using large width, this finding suggests an interesting tradeoff between computational cost and statistical complexity.
> Moreover, although the large width is one main drawback of the current mean-field analysis, as shown in Table 1, our result already represents a noticeable improvement over existing mean-field analyses, in terms of both the width (prior results required $N=\mathcal{O}(d^d)$) and the iteration complexity (prior works assumed $t\to\infty$).
> Therefore, we believe that our analysis serves as an important step toward more efficient (quantitative) learning guarantees for mean-field neural networks.
>
>
> **Q (minor):** *I glanced at the code for the experiments and noticed that $m = 2,000$.*
> **A:** Thank you for the close reading. This is because we also conducted experiments for $k=3,4$ parity problems in the Appendix in which we employed $m=2,000$ and we uploaded the code for that setting. Although only a few lines differ for each experiment, we will include all three experiments in the new supplementary file.
>
> We would be happy to clarify any concerns or answer any questions that may come up during the discussion period.

---

> > ### Comment · Reviewer_1qsa · 2023-08-14
> >
> > I would like to thank the authors for their point-by-point rebuttals to my (perhaps superficial) concerns. I have also carefully looked over the reviews provided by other reviewers.
> >
> > I  have increased my score to 6 in light of the rebuttal + other reviews as they have helped me gain a better understanding of the paper.

---

### Official Review · Reviewer_afRb · 2023-07-09

**Soundness:** 3 good
**Presentation:** 2 fair
**Contribution:** 3 good
**Rating:** 6
**Confidence:** 3

**Summary:**

This paper proves optimization and generalization guarantees for MFLD for the $k$-sparse parity problem. It proves that if the network is sufficiently overparameterized (width > $e^{\Omega(d)}$) then $n = d$ samples suffice, which is independent of $k$. Furthermore it proves an exponential convergence rate when $n \ge d^2$.

**Strengths:**

- The paper is able to prove concrete Rademacher-based generalization guarantees for MFLD, including explicit $\epsilon$ dependencies in different regimes

Also, I think the comparisons with Chen and Meka (2020) are unnecessary as that paper focuses on the case when the covariates are Gaussian which is a very different problem.

**Weaknesses:**

- This paper does not consider label noise which would likely break the exponential convergence rate in the $n \ge d^2$ regime. It is not clear to me whether the techniques would extend to this setting.
- Like many mean field papers, discretizing the dynamics requires exponential width (at least $e^{\Omega(d)}$). The tradeoff is therefore width $O(1)$ and sample complexity $O(d^k)$ vs width $e^{O(d)}$ and sample complexity $d$. However, this problem is not unique to this paper and appears fundamental to a lot of mean field analysis.

**Questions:**

- How difficult would it be to adapt the techniques in this paper to the case of label noise (i.e. flip each label with probability $p < 1/2$)?

**Limitations:**

The authors have adequately addressed the limitations of this paper.

---

> ### Author Rebuttal · Authors · 2023-08-09
>
> Thank you for your supportive comments. We address the technical comments below.
>
> **Q:** *How difficult would it be to adapt the techniques in this paper to the case of label noise (i.e. flip each label with probability $p < 1/2$)?*
> **A:** We believe that extending our result to a situation with label flipping is not difficult and the almost same results would hold (i.e., the exponential convergence of the classification error for $d^2 < n$ and $d/n$ classification error for $d < n$).
> However, we need to take care of non-monotonicity of the conditional expectation of the loss $f \mapsto \mathbb{E}_Y[\ell(Yf)|X]$ when there is a label noise. This requires additional technical difficulty, but we think it can be resolved by using a small $\bar{R}$.
>
> Indeed, the exponential convergence of the classification error is shown by [R1] for $p < 1/2$ label flipping in kernel classification. The essential point of the proof is to show the $L^\infty$-convergence of the classifier, which is what we have shown in our analysis. Hence, we believe that our analysis would be a good starting point to explore more difficult and realistic settings.
>
> [R1] Koltchinskii, V., Beznosova, O. (2005). Exponential Convergence Rates in Classification. In: Auer, P., Meir, R. (eds) Learning Theory. COLT 2005. Lecture Notes in Computer Science, vol 3559, pp.295--307, 2005.
>
>
> **Q:** *Exponentially large width.*
> **A:** Indeed this is one main drawback of the current mean-field analysis. This being said, as shown in Table 1, our result already represents a noticeable improvement over existing mean-field analyses, in terms of both the width (prior results required $N=\mathcal{O}(d^d)$) and the iteration complexity (prior works assumed $t\to\infty$), even though our results can handle the more general $k$-parity setting.
> Therefore, we believe that our analysis serves as an important step toward more efficient (quantitative) learning guarantees for mean-field neural networks.
>
> We would be happy to clarify any concerns or answer any questions that may come up during the discussion period.

---

> > ### Comment · Reviewer_afRb · 2023-08-15
> >
> > Thank you for the clarifications. I have decided to keep my score.

---

### Decision · Program_Chairs · 2023-09-21

**Decision:**

Accept (poster)

**Comment:**

The reviewers unanimously find the paper interesting and result worth publication. The area chair agrees with the evaluation after reading the discussion and the manuscript.